# The alphaherpesvirus conserved pUS10 is important for natural infection and its expression is regulated by the conserved *Herpesviridae* protein kinase (CHPK)

**Nagendraprabhu Ponnuraj**[1], **Haji Akbar**[1], **Justine V. Arrington**[2], **Stephen J. Spatz**[3], **Balaji Nagarajan**[4,5], **Umesh R. Desai**[5], **Keith W. Jarosinski**[1] *

1 Department of Pathobiology, College of Veterinary Medicine, University of Illinois at Urbana-Champaign, Urbana, Illinois, United States of America, 2 Protein Sciences Facility, Roy J. Carver Biotechnology Center, University of Illinois Urbana-Champaign, Urbana, Illinois, United States of America, 3 US National Poultry Research Laboratory, Agricultural Research Service, United States Department of Agriculture, Athens, Georgia, United States of America, 4 Department of Medicinal Chemistry, School of Pharmacy, Virginia Commonwealth University, Richmond, Virginia, United States of America, 5 Institute for Structural Biology, Drug Discovery and Development, Virginia Commonwealth University, Richmond, Virginia, United States of America

* kj4@illinois.edu

**Data Availability Statement:** All relevant data are within the manuscript and its supporting information files.

## Abstract

Conserved *Herpesviridae* protein kinases (CHPK) are conserved among all members of the *Herpesviridae*. Herpesviruses lacking CHPK propagate in cell culture at varying degrees, depending on the virus and cell culture system. CHPK is dispensable for Marek's disease herpesvirus (MDV) replication in cell culture and experimental infection in chickens; however, CHPK—particularly its kinase activity—is essential for horizontal transmission in chickens, also known as natural infection. To address the importance of CHPK during natural infection in chickens, we used liquid chromatography-tandem mass spectrometry (LC-MS/MS) based proteomics of samples collected from live chickens. Comparing modification of viral proteins in feather follicle epithelial (FFE) cells infected with wildtype or a CHPK-null virus, we identified the US10 protein (pUS10) as a potential target for CHPK *in vivo*. When expression of pUS10 was evaluated in cell culture and in FFE skin cells during *in vivo* infection, pUS10 was severely reduced or abrogated in cells infected with CHPK mutant or CHPK-null viruses, respectively, indicating a potential role for pUS10 in transmission. To test this hypothesis, US10 was deleted from the MDV genome, and the reconstituted virus was tested for replication, horizontal transmission, and disease induction. Our results showed that removal of US10 had no effect on the ability of MDV to transmit in experimentally infected chickens, but disease induction in naturally infected chickens was significantly reduced. These results show CHPK is necessary for pUS10 expression both in cell culture and in the host, and pUS10 is important for disease induction during natural infection.

**Funding:** This report was supported by Agriculture and Food Research Initiative Competitive Grant nos. 2016-67015-26777 and 2020-67015-21399 from the USDA National Institute of Food and Agriculture to KWJ; USDA-ARS NACA agreement nos. 58-6040-8-037 and 58-6040-0-015 to KWJ and SJS; and the National Institutes of Health grants U01 CA241951 and P01 HL151333 to URD. The funders had no role in study design, data collection and analysis, decision to publish, or preparation of the manuscript. The authors NP, HA, JVA, and BN received no specific funding for this work.

**Competing interests:** The authors have declared there is no competing interest.

## Author summary

Marek's disease herpesvirus (MDV) is an important pathogen in the poultry industry. Current vaccines reduce disease but do not protect chickens from infection. Understanding natural infection provides vital information for developing potential therapies to protect against Marek's disease, while also providing a robust natural virus-host model to study herpesvirus pathogenesis. The conserved *Herpesviridae* protein kinase (CHPK) of MDV is essential for the initiation of natural infection, and we identified the virion protein US10 as a potential target for CHPK during natural infection. Our results showed that MDV CHPK was required for expression of pUS10 in both cell culture and in chickens. Although MDV lacking pUS10 replicated and spread in experimentally infected chickens, it was less virulent implicating an important role of pUS10 in natural infection of the host. These results emphasize the importance of studying specific viral proteins and their functions during infection of the natural host.

## Introduction

Herpesviruses infect most animal species studied, including birds, turtles, reptiles, fish, and even oysters, and are highly adapted to their host species, having coevolved for millions of years. Despite having a broad host range and pathogenic potentials, all herpesviruses share a core set of genes. This includes the conserved *Herpesviridae* protein kinase (CHPK) that is common to all members of the *Herpesviridae* and produces serine/threonine kinases [1]. The extent of this conservation suggests important roles for CHPKs in the herpesvirus replication cycle, although there is a considerable amount of sequence divergence among them overall between the three Alpha-, Beta-, and Gamma-herpesvirus subfamilies [2]. CHPK orthologues are incorporated as part of the tegument [3–5], suggesting an important role during virus entry and initiation of infection. Most CHPK-mutant or -null viruses tested are viable using traditional cell and tissue culture systems [6–10]. CHPKs have been linked to multiple processes during replication including nuclear egress [11–17], tegument association/dissociation [18, 19], viral gene expression [8, 12, 20–23], viral DNA replication [12, 16, 17, 24–29], DNA damage responses [30, 31], cell-cycle regulation [32–35], and evasion of the interferon response [36–38].

During natural infection of chickens with *Gallid alphaherpesvirus* 2 (GaHV2), better known as Marek's disease herpesvirus (MDV), CHPK is essential for infection [23, 39, 40]. Natural infection of MDV begins through the respiratory route where cytolytic infection initiates in B and T lymphocytes and then disseminates to immunologic organs in the chicken, causing immune suppression and transformation of T cells into malignant cancer. However, to spread from host-to-host, infection of feather follicle (FF) epithelial (FFE) cells in the skin is required to produce infectious particles that are shed into the environment. This provides a continuous source of infectious virus in the poultry house. This method of transmission is similar to human chicken pox infections with varicella zoster herpesvirus (VZV) [41].

As with other members of the *Herpesviridae* studied, MDV CHPK is dispensable for virus replication in tissue culture cells [2, 39, 40, 42] and chickens during experimental infection. However, it is required for transmission, suggesting a critical role for this kinase during natural infection. Recent work in our laboratory showed that expression, localization, and stability of MDV CHPK and expression of late viral genes are severely affected in the absence of CHPK activity [23]. However, to date, no specific cellular or viral protein substrates have been identified for MDV CHPK. All CHPK orthologues previously studied are capable of

autophosphorylation, and a few cellular and viral protein targets have been identified depending on the herpesvirus studied [1, 43]. Thus far, only two proteins, EF-1δ [44–47] and Hsp90 [48] are commonly phosphorylated by CHPKs.

US10 is a virion protein associated with the tegument and is conserved among the alpha-herpesviruses, including the prototype Herpes simplex virus 1 (HSV-1) [49]. US10 homologs of *Equid herpesvirus* 1 (EHV-1), HSV-1, and VZV possess the consensus CCHC-type (C-X3-C-X3-H-X3-C) zinc finger domain [49–52], but the functional significance of this domain is not known at this time. MDV US10 is located within the unique short (US) region of the MDV genome, but, interestingly, it is transposed relative to other US genes in the region when compared to HSV-1 [53]. US10 is dispensable for cell culture replication of HSV-1 [54, 55]. Similarly, Parcells et al. [56] showed that US10 of MDV, as part of a large deleted cluster of genes, was also dispensable for cell culture replication and nonessential for replication in experimentally infected chickens. Transmission was not observed in these experiments, suggesting that one or more of the genes included in this cluster deletion may be important for transmission. In another report, direct insertional mutagenesis (Lac Z) of the MDV US10 loci resulted in the loss of transmission, suggesting a role of US10 for transmission [57]. However, this study did not test transmission of the parental virus, making it difficult to conclusively determine whether the lack of US10 was directly responsible for the lack of transmission.

Here, we used liquid chromatography and tandem mass spectrometry (LC-MS/MS) based proteomics to identify potential substrates for MDV CHPK in the host. We identified pUS10 as a downstream target of CHPK activity and showed that pUS10 expression is dependent on CHPK in both cell culture and *in vivo*. The data combined suggested US10 expression was dependent on CHPK, linking US10 to natural infection. Using a US10-null virus, we conclusively demonstrated that while pUS10 is not essential for transmission of MDV, it is a virulence factor and influences the development of disease during natural infection.

## Results

### MDV CHPK is important for infectivity

Former studies in our laboratory have shown that MDV CHPK is dispensable for virus replication in cell culture and chickens experimentally infected with recombinant (r) MDV but is required for transmission in chickens [40]. Interestingly, there does not appear to be any defect in the ability of MDV lacking CHPK activity to reach the FFE skin cells, nor is there any obvious defect in the assembly of virions in FFE skin cells [23]. However, the infectivity of these viral particles was never examined in our former study. To address this, additional samples were collected from chickens infected with (v)CHPKwt or vCHPKmut from our previous experiment, and skin/feather samples were collected to examine the structure of virions within FFE skin cells and their infectivity. Consistent with previous studies, expression of the late pUL47eGFP protein could be observed in FFE skin cells infected with vCHPKwt or vCHPKmut (Fig 1A). Transmission electron microscopy (TEM) showed that virions are produced in both vCHPKwt and vCHPKmut-infected FFE skin cells and there does not appear to be any obvious structural defects, although quantitative data was not possible.

Since there were no visual defects in virions between MDV expressing wildtype CHPK protein (pCHPKwt) or mutant CHPK protein (pCHPKmut) that could explain the lack of horizontal transmission, we next focused on examining infection of purified virions in FFE skin cells. Although virions were present in vCHPKmut-infected FFE skin cells (Fig 1A), these virions may lack infectivity. To test this, MD virions were extracted from the skins/feathers of vCHPKwt and vCHPKmut-infected chickens at 21 days post-infection (pi) using a modified Ficoll gradient protocol (Fig 1B) and used for immunoblotting (Fig 1C) and infectivity studies (Fig 1D). Expression

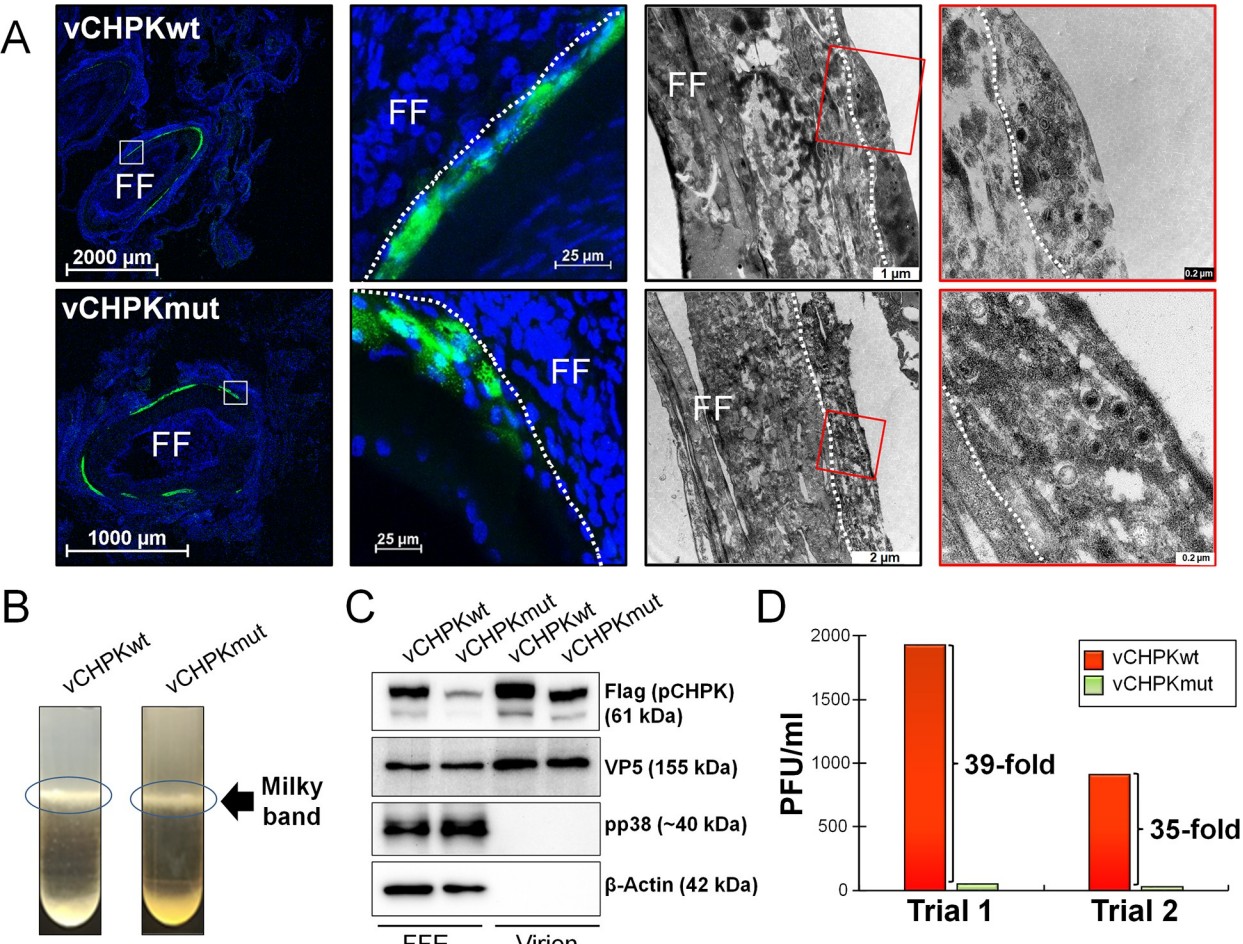

**Fig 1. CHPK is required for efficient infection.** (A) Infected FFE skin cells were isolated using fluorescent protein expression to identify vCHPKwt and vCHPKmut-infected cells. TEM was used to examine viral particles in representative skin epithelial cells of vCHPKwt- and vCHPKmut-infected cells. Regions highlighted in red were blown up to better visualize viral particles. White dotted lines were used to delineate the epithelial skin layer (FFE cells). (B) Viral particles were purified from the infected skins of three birds per group, combined, and the "milky band" containing enveloped viral particles was isolated for both groups. (C and D) Purified viruses from (B) were used for immunoblotting (C) and infectivity assays (D) on CKCs, and plaques were counted 5 days pi in two trials.

of pCHPK in MDV virions was present in both vCHPKwt and vCHPKmut samples in FFE skin cells, and both were present in virion preparations (Fig 1C). Consistent with previous results, the relative level of CHPKmut was lower compared to CHPKwt under reducing conditions [23]. Immunoblotting for MDV pp38 and cellular β-Actin, both expected to be absent in MD virions, were used as controls for virion extract purity (Fig 1C) and both were absent in the virion preparation. The relative amount of plaques produced from the virion preparations by vCHPKmut was >35-fold lower compared to vCHPKwt samples (Fig 1D). These results, along with our former data, show that the relative amount of MD virions in FFE skin samples produced by vCHPKmut is similar to vCHPKwt and both CHPKs are present in the virions; however, the virions collected from vCHPKmut are severely defective in infectivity.

## Phosphorylation of pUS10 in FFE skin cells infected with rMDVs

Since virions collected from vCHPKmut-infected chickens were severely defective in infectivity and CHPK activity is essential for transmission [23, 40], we directed our attention to

potential substrates for MDV pCHPK in FFE skin cells using LC-MS/MS-based phosphoproteomics. The analysis of protein extracts from feather follicles collected between 21–28 days following inoculation of birds with vUL47eGFP, vCHPKwt, or vCHPKmut [23] revealed a unique peptide [R].VD**S**PKEQSYDILSAGGEHVALLPK.[S] belonging to pUS10 that was consistently phosphorylated on Serine 20 (S20) in vUL47eGFP and vCHPKwt-infected cells but not in vCHPKmut-infected FFE cells. Furthermore, the sequence containing the phosphorylation site of interest (**S**PKE) was identified as a casein kinase 2 (CKII) (SxxE/D) and cyclin-dependent kinase (CDK) (SPxK) motifs using MyHits [58] and ScanSite 4.0 [59] motif scans, in agreement with potential targets reported for other CHPKs [60–62]. Therefore, we focused our subsequent studies on pUS10.

## Generation of pCHPK-null and epitope tagged pUS10 rMDV

Since we have previously shown that a single amino acid change can result in reversion during MDV replication in chickens [39, 63], we generated a pCHPK-null rMDV (rΔCHPK) in which much of the UL13 gene (aa 116–496) was deleted (Fig 2). Plaque size assays of reconstituted vΔCHPK in chick embryo cells (CECs) showed no significant difference between it and vCHPKwt or vCHPKmut (Fig 3A). In addition, since commercial antibodies against MDV pUS10 are unavailable, we tagged the C-terminus of pUS10 with 2×HA epitope in the genomes of parental vCHPKwt, vCHPKmut [23], and the newly generated vΔCHPK. All three of the recombinant viruses also express pUL47eGFP [64].

Fig 2A shows a schematic representation of the MDV genome and rMDV generated including rCHPKwt/10HA, rCHPKmut/10HA, and rΔCHPK/10HA. Restriction fragment length polymorphism (RFLP) analysis of each parental and derivatives are shown in Fig 2B to confirm the integrity of the bacterial artificial chromosome (BAC) clones. PCR and DNA sequencing of each gene was used to further confirm that each clone was correct at the nucleotide level (S1–S3 Figs) using previously described primers [40].

## Analysis of pUS10 expression in cell culture

Plaque size assays showed no effect on the size of induced plaques with the 2×HA tag on pUS10 relative to those without (Fig 3A), indicating the C-terminal 2×HA tag on pUS10 had no effect on MDV cell culture replication. Next, we determined whether the pUS10c2×HA protein could be detected in MDV-infected CECs using indirect immunofluorescence assays (IFAs) and immunoblotting. Using an anti-HA antibody, pUS10 expression could be readily detected in vCHPKwt/10HA-infected cells (Fig 3B). However, the expression of pUS10 was lower in vCHPKmut/10HA-infected cells and undetectable in vΔCHPK/10HA-infected cells. To confirm these differences, confocal microscopy was used (Fig 3C). Consistent with our former results on pCHPK expression [23], pCHPKwt was abundantly expressed in both the cytoplasms and nuclei, while pCHPKmut was more predominantly expressed in the nuclei of infected cells. Interestingly, the pΔCHPK was almost exclusively located in the nuclei in vΔCHPK/10HA-infected cells. Expression of pUS10 was abundantly detected in vCHPKwt/10HA-infected cells, while expression was visually reduced in cells infected with vCHPKmut/10HA, and undetectable in vΔCHPK/10HA-infected cells. Next, pCHPK and pUS10 expression was visualized using immunoblotting (Fig 3D). Consistent with data for IFAs, immunoblotting showed low pUS10 expression in cells infected with vCHPKmut/10HA and virtually no pUS10 expression in vΔCHPK/10HA-infected cells. Following this result, we sequenced US10 using DNA obtained from infected CECs and confirmed the C-terminal 2×HA tag was still present in the viral genomes.

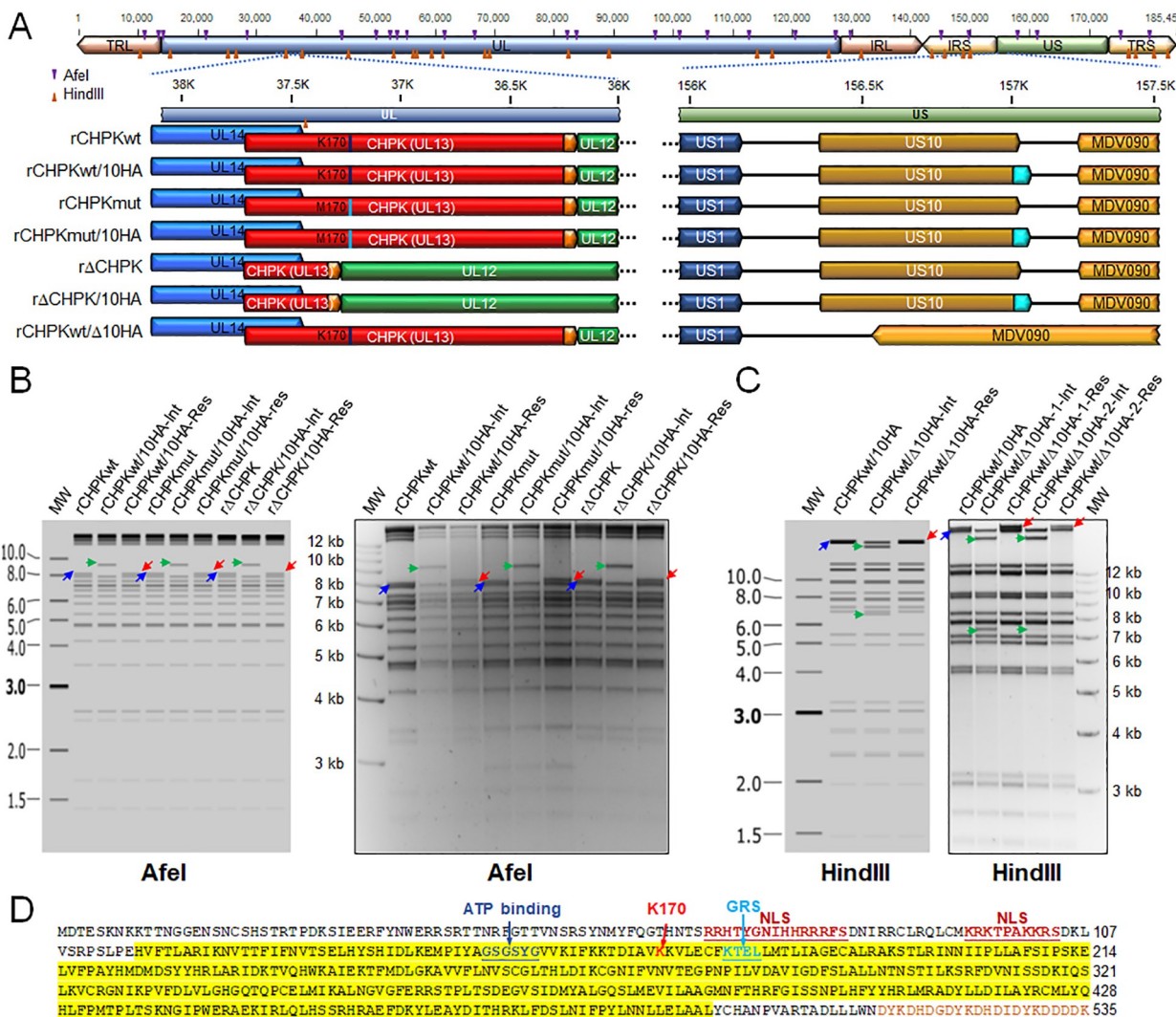

**Fig 2. Generation of rMDVs.** (A) Schematic representation of the MDV genome depicting the locations of the terminal repeat long (TRL) and short (TRS), internal repeat short (IRS), and unique long (UL) and short (US) regions. A portion of the UL and US regions are expanded to show the relevant genes including UL12, UL13 (CHPK), and UL14 in the UL region and US1, US10, and MDV090 in the US region. Differences between the rMDVs are shown, including the amino acid located at position 170 in pCHPK and the presence of the 3×Flag and 2×HA epitopes on pCHPK and pUS10, respectively. (B) Predicted and actual RFLP analysis of rMDV CHPK and US10 BAC clones. BAC DNA obtained for each parental (rCHPKwt, rCHPKmut, and rΔCHPK), integrates (rCHPKwt/10HA-Int, rCHPKmut/10HA-Int, and rΔCHPK/10HA-Int), and resolved clone (rCHPKwt/10HA-Res, rCHPKmut/10HA-Res, and rΔCHPK/10HA-Res) were digested with AfeI and examined using RFLP analysis. Insertion of the I-*SceI-aphAI* +2×HA cassette into the 8,055 bp fragment (right directed blue arrow) results in a 9,195 bp fragment (right directed green arrow) during integration. This fragment is then reduced to 8,121 bp fragment (left directed red arrow) during resolution and removal of I-*SceI-aphAI* cassette. (C) Predicted and actual RFLP analysis of rMDV ΔUS10 BAC clones. BAC DNA obtained for rCHPKwt/10HA, rCHPKwt/Δ10HA-Int, and rCHPKwt/Δ10HA-Res were digested with HindIII and examined using RFLP analysis. Integration of the I-*SceI-aphAI* cassette into 26,782 bp fragment (right directed blue arrow) containing US10c2×HA inserts a HindIII site creating two fragments, 20,505 and 6,600 bp fragments (right directed green arrows). Resolution removes the I-*SceI-aphAI* cassette and the additional HindIII site creating a 26,077 bp fragment (left directed red arrow). The molecular weight marker used was the GeneRuler 1 kb Plus DNA Ladder from Thermo Scientific (Carlsbad, CA). No extraneous alterations are evident. (D) MDV pCHPK with C-terminal 3×Flag tag (brown letters). Yellow highlighted area indicates protein sequence deleted in vΔCHPK/10HA. The invariant lysine (K170) and ATP binding sites are noted [96, 97]. Putative nuclear localization signals (NLS) and Golgi retention sequence (GRS) were identified based on previous studies [98–100] or predicted using NLStradamus [101] and MyHits [102].

Although pUS10 was barely detectable in CECs infected with vCHPKmut and vΔCHPK, its expression was (i) abundant when pCHPKwt was co-expressed with pUS10 and (ii) both

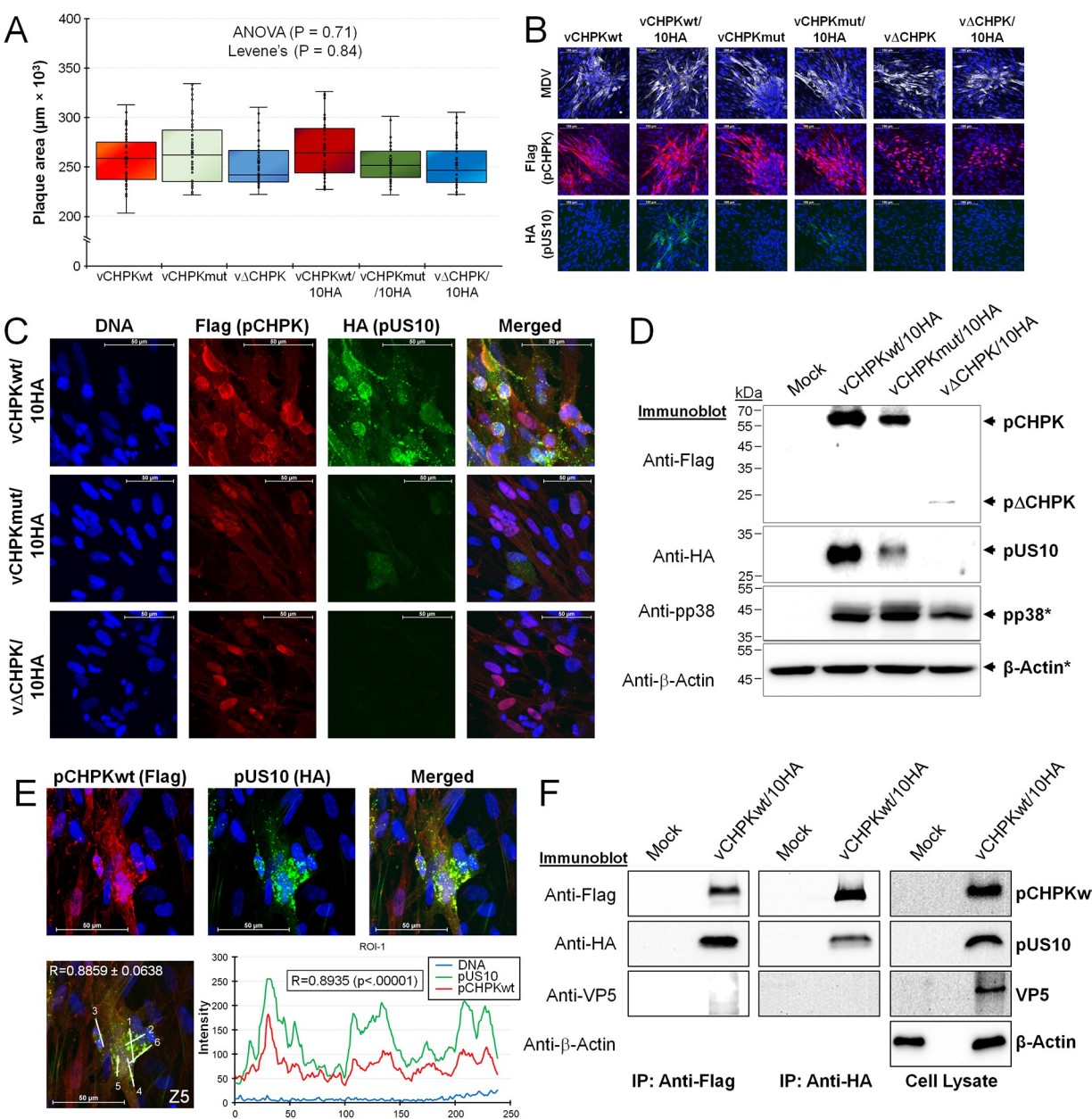

**Fig 3. Replication of rMDVs in tissue culture cells and expression of pCHPK and pUS10.** (A) Plaque areas were measured for rMDVs reconstituted from each parental virus (vCHPKwt, vCHPKmut, vΔCHPK) and HA-tagged US10 virus (vCHPKwt/10HA, vCHPKmut/10HA, vΔCHPK/10HA). There were no significant differences between all groups using One-way ANOVA ($P$ = 0.71, n = 300) and Levene's Test ($P$ = 0.84, n = 300). (B) Representative plaques induced by each virus stained with anti-MDV, -Flag, and -HA antibodies. Included in each image is Hoechst 33342 staining of nuclei for reference. Scale bars are 100 μm. (C) Confocal microscopy of vCHPKwt/10HA-, vCHPKmut/10HA-, or vΔCHPK/10HA-infected CECs. Cells were stained for Flag (CHPK) and HA (US10), and DNA (Hoechst 33342). Scale bars are 50 μm. (D) Immunoblotting of total protein collected from mock-infected CECs or CECs infected with vCHPKwt/10HA, vCHPKmut/10HA, or vΔCHPK/10HA. To examine CHPK and US10 expression, proteins were electrophoresed under non-reducing conditions and blotted with anti-Flag (CHPK) and -HA (US10) antibodies. Protein was also run under reducing conditions (*) and blotted with anti-β-Actin and -MDV pp38 as protein loading and infection level controls, respectively. (E) Confocal images of CECs infected with vCHPKwt/10HA were used to quantify the colocalization of pCHPK and pUS10. A representative image of a plaque stained with anti-Flag (pCHPKwt) and -HA (pUS10) and merged. Hoechst was used to identify cell nuclei. Scale bars are 50 μm. A single z-section with linear ROIs labeled and the profile plot for ROI-1, and its Pearson Correlation Coefficient R is shown. The average Pearson Correlation Coefficient R between pCHPKwt and pUS10 was 0.8859 ± 0.0638. (F) Uninfected CECs (Mock) or CECs infected with vCHPKwt/10HA were used for protein extraction and total cell lysate was immunoblotted for pCHPKwt (Flag), pUS10 (HA), VP5 (non-specific control), and β-Actin (protein loading control). Total lysate was immunoprecipitated with anti-Flag Fab or anti-HA combined with protein A/G-beads, electrophoresed on an SDS-PAGE gel, and then immunoblotted using anti-Flag, -HA, or VP5 antibodies.

proteins appeared to colocalize in cells. To confirm this, quantitative analysis of colocalization was performed in vCHPKwt/10HA-infected CECs and showed that pCHPKwt and pUS10 were positively correlated with a mean Pearson Correlation Coefficient R value of 0.8859 ± 0.0638 obtained with five regions of interests (ROIs) from three individual cells showing roughly equal expression of pCHPK and pUS10. A represented cell and ROIs are shown in Fig 3E. To determine whether this interaction was specific, total cell lysates from vCHPKwt/10HA-infected CECs were used in co-immunoprecipitation assays (Fig 3F). Consistent with the colocalization data (Fig 3E), pCHPKwt and pUS10HA interacted, but neither protein interacted with the VP5 capsid protein we surmised would not interact based on former studies [1, 43]. These results show pCHPK and pUS10 interact during replication in cell culture and pUS10 expression is dependent on functional pCHPK.

Next, we examined the localization of pCHPK and pUS10 out of the context of infection using transfection with expression plasmids (designated pc) in cell culture. Consistent with viral infection, pCHPKwt and pCHPKmut were diffusely located in the cytoplasm and nucleus, while pΔCHPK was mostly nuclear (Fig 4A). When both pcCHPK and pcUS10eGFP were co-transfected, there was clear colocalization of pUS10 with pCHPKwt and pCHPKmut, while pΔCHPK and pUS10 were not colocalized (Fig 4B). Quantitative analysis of colocalization showed a positive correlation with mean R values of 0.93 ± 0.04 between pUS10 and pCHPKwt and 0.88 ± 0.09 between pUS10 with pCHPKmut, while localization of pΔCHPK and pUS10 was not correlated with a mean R value of -0.08 ± 0.23 (Fig 4D). These data confirm colocalization of pCHPK and pUS10 is not dependent on viral infection and likely, the region in which pCHPK and pUS10 interact is in the region deleted in pΔCHPK (aa 116–496).

## Analysis of pCHPK and pUS10 stability

Since expression of pUS10 was reduced or abrogated in CECs infected with vCHPKmut/10HA or vΔCHPK/10HA, respectively, RT-qPCR assays and immunoblotting were used to address why pUS10 levels were altered. First, we measured the level of US10 and UL13 transcripts using RT-qPCR assays to determine whether the reduced protein expression was at the transcriptional level. There were no differences (P >0.05) in both UL13 and US10 transcripts during replication in CECs (Fig 5A), suggesting the cause of the significantly decreased levels of pUS10 is post-transcriptional. We also included US1, US3, UL47, and UL49 mRNA levels and found no differences in viral transcription of these genes as well, suggesting pCHPK plays no role in viral gene transcription.

Since mRNA levels of both UL13 and US10 are unaffected during cell culture propagation, this suggested both proteins may be targeted for degradation in the vCHPKmut/10HA and vΔCHPK/10HA mutant viruses. This would be consistent with our previous work showing pCHPKmut was structurally unstable and possibly targeted for degradation [23]. To test this, infected cells were treated with DMSO (vehicle control) or MG132 which inhibits the proteasome [65]. Fig 5B suggested pCHPKwt levels remained unchanged with MG132 treatment, while pCHPKmut and pΔCHPK levels were increased moderately following MG132 treatment. However, pUS10 levels were unaffected or slightly decreased with MG132 treatment. To confirm this observation, the average protein levels were quantified using densitometry in three independent experiments (Fig 5C). Both pCHPKmut and pΔCHPK protein levels were significantly lower than pCHPKwt, confirming our initial observations (Fig 3D) and former studies [23]. Inhibition of the proteasome increased levels of pCHPKmut, while there were no changes to protein levels of pCHPKwt or pΔCHPK (Fig 5C). With respect to pUS10 expression, there were significant decreases in pUS10 protein levels between CECs infected with vCHPKwt/10HA compared vCHPKmut/10HA, vCHPKwt/10HA compared to vΔCHPK/10HA, and

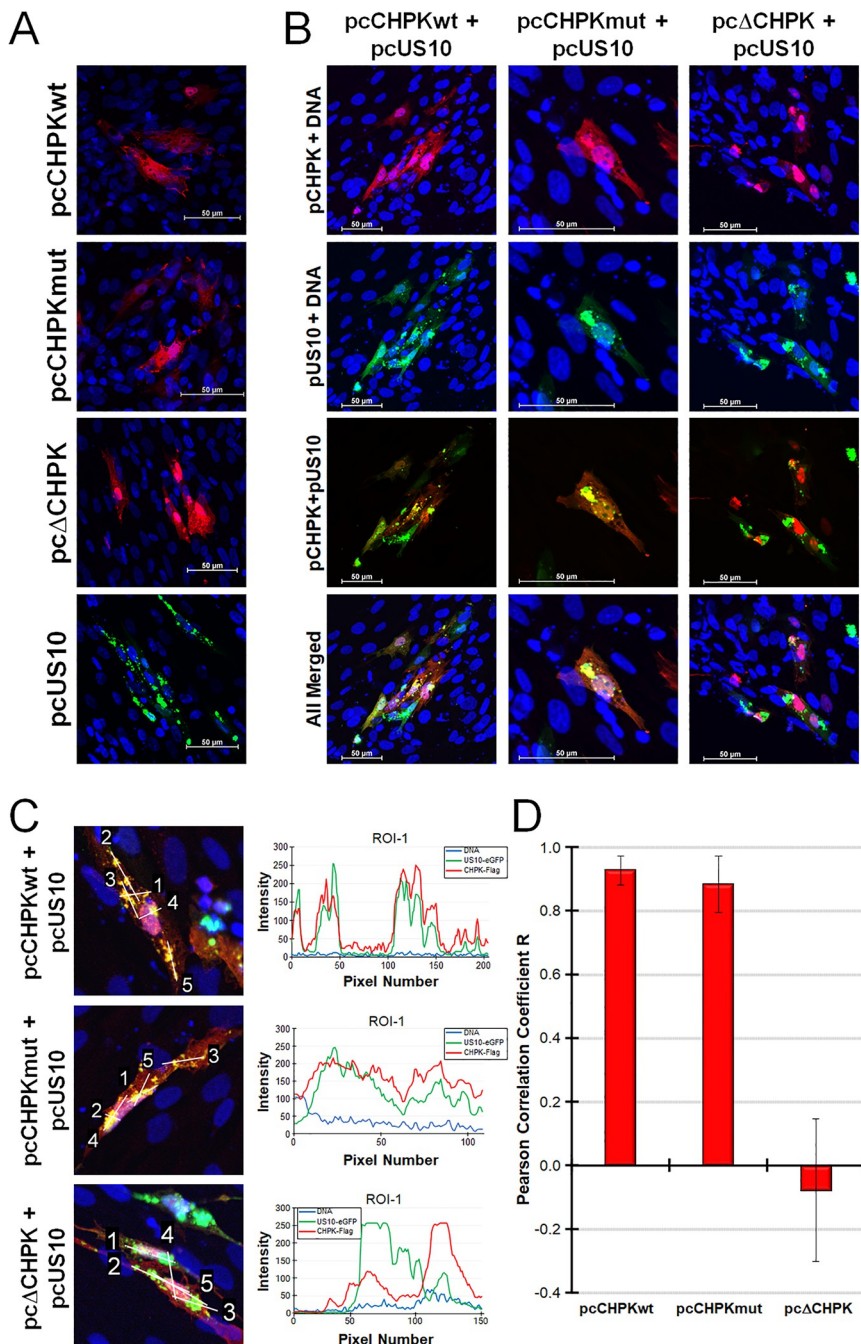

**Fig 4. pCHPK and pUS10 colocalize in cells.** (A) CECs were transfected with pcCHPKwt, pcCHPKmut, pcΔCHPK, or pcUS10eGFP and 24 h, fixed and probed with anti-Flag antibody to detect pCHPK or visualized directly for pUS10eGFP. Images of 10–15 z-sections were compiled into a single image using Adobe Photoshop. (B) CECs were co-transfected with pcCHPK expression constructs plus pcUS10eGFP and fixed 24 h later and probed with anti-Flag antibody. Images of 10–15 z-sections were compiled into a single image using Adobe Photoshop. (C) Representative images of single z-sections with linear ROIs labeled and the profile plots for ROI-1 (to the right of each image) were used for quantitative analysis of colocalization. (D) Graphical presentation of the mean Pearson Correlation Coefficient R between pcUS10eGFP and the pcCHPK derivates. Scale bars are 50 μm.

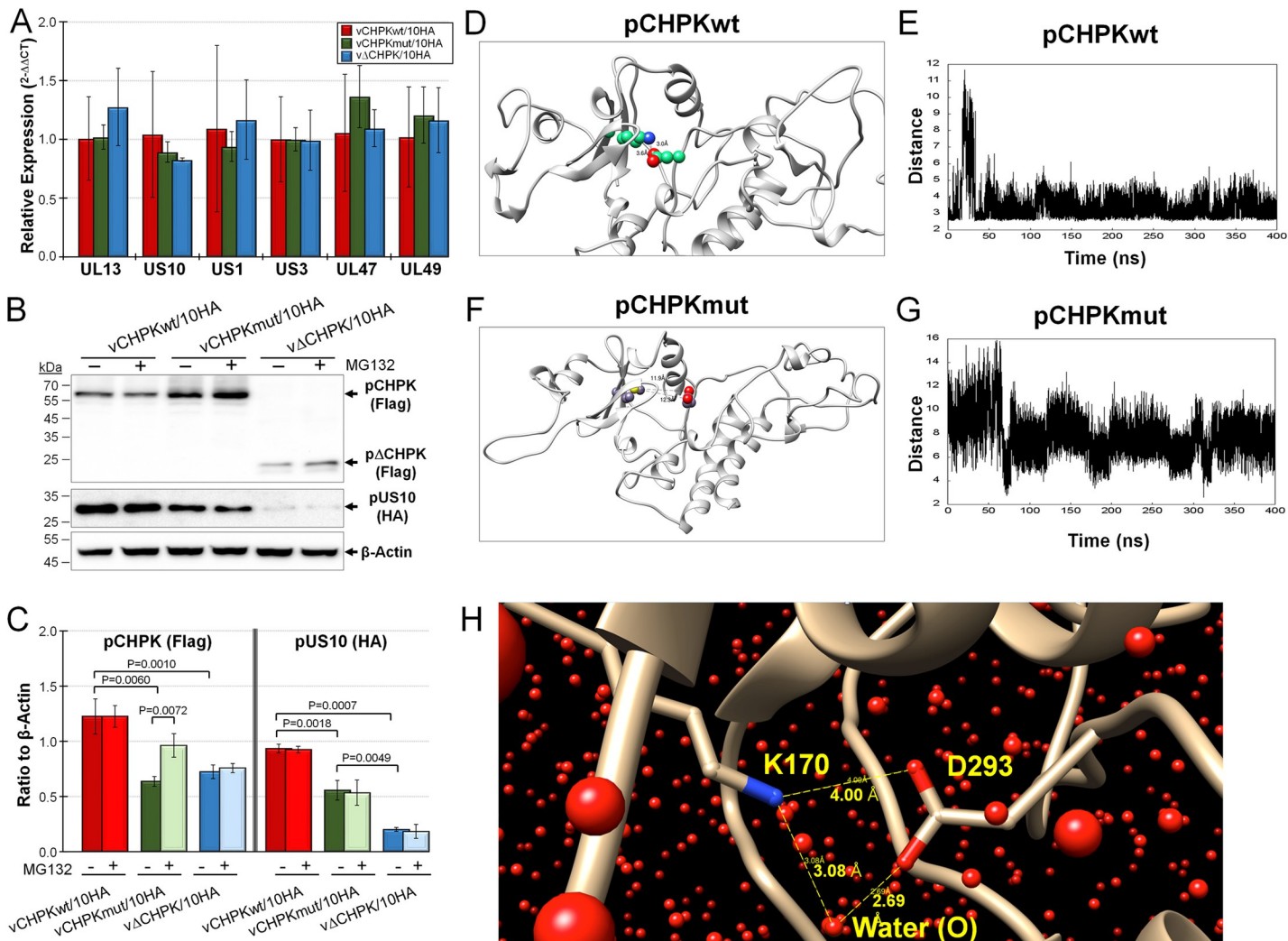

**Fig 5. pCHPK and pUS10 protein stability.** (A) Relative expression of UL13, US10, US1, UL47, and UL49 mRNA in vCHPKwt/10HA-, vCHPKmut/10HA-, and vΔCHPK/10HA-infected CECs after 5 days pi. Means of $2^{-\Delta\Delta CT}$ for each gene are shown with standard deviations relative to viral gene expression in vCHPKwt/10HA. There were no differences between all three viruses for all genes. (B) CECs infected with vCHPKwt/10HA, vCHPKmut/10HA, or vΔCHPK/10HA were infected for 3 days, then treated with DMSO or MG132 for 24 h to block proteasomal-mediated degradation. Expression of CHPK (Flag), US10 (HA), and β-Actin were examined using immunoblotting. (C) The average ratio of pCHPK (Flag) and pUS10 (HA) protein to β-Actin in three independent experiments with and without MG132 treatment. Statistical differences are shown with P values using Student's *t* tests. (D) A representative frame from the MDs trajectory of the pCHPKwt (K170) showing the ionic bond between residues K170 and D293, which helps maintain its stable 3D form. (E) Variation in the ionic bond length (distance between K170 and D293) as a function of the MDs simulation time. The average distance between the two residues was found to be 3.45 Å. (F) A representative frame from the MDs trajectory of pCHPKmut (M170) showing loss of ionic bond with D293. (G) Variation in the distance between M170 and D293 as a function of the MDs simulation time. The average distance was calculated to be 7.82 Å. Notice the wide fluctuations between the two residues suggesting high conformational flexibility and dynamism of the mutant protein. (H) Water-mediated bridging interaction between the K170 and D293 residues, apart from the cationic and anionic direct interaction in the native pCHPK, also stabilized the overall 3D structure of the wildtype protein. Bridged water molecules were present in ~66% of MDs time frames. Shown above are the interacting residues and the oxygen atoms of the water molecules. Also shown are the distances between donor and acceptor atoms of interacting residues and water labeled (O) (yellow dashed lines with distances).

vCHPKmut/10HA compared to vΔCHPK/10HA, confirming our initial IFA (Fig 3B and 3C) and immunoblotting (Figs 3D and 5B) results. MG132 had no effect on pUS10 protein levels.

Next, we studied the conformational and dynamic properties of wildtype and mutant pCHPKs, pCHPKwt (K170) and pCHPKmut (M170), respectively, in the presence of explicit water using molecular dynamics (MDs). As the crystal structure of neither protein was available, we modeled the two structures from their primary sequences using Swiss-Model engine

(see Materials and Methods) and Alpha Fold2 (https://alphafold.ebi.ac.uk), which gave an identical structure (S4 Fig). The 400 ns long MDs showed that the pCHPKwt (K170) protein forms a strong ionic interaction between the wildtype K170 and D293, as shown in Fig 5D. Over the MDs time frame, this salt bridge maintained an average distance of 3.45 Å, which indicates its role in maintaining the local three-dimensional structure (Fig 5E). Thus, this salt bridge adds considerable stability to the protein structure. In addition, the contributing residues are further engaged through interactions with water molecules that bridge the two residues (Fig 5H). An occupancy of 66% was calculated for waters bridging K170 and D293 over MDs simulation.

In contrast, the mutant protein carrying M170 failed to form the ionic bond with D293 (or with any other neighboring residue) as one would predict based on the nature of the side chain of the methionine residue. Importantly, the absence of the wildtype K170—D293 ionic bond in pCHPKmut (M170) led to significant dynamic mobility as evidenced by the nearly doubling of the average distance (7.82 Å) between the two residues (Fig 5G). Further, considerable gyrational motions were observed over the MDs time frame signifying the flexibility arising from the loss of the ionic interaction. It is very likely that this mobility or dynamism in the 3D structure of the mutant could measurably reduce the activity and stability of the protein. These data combined show the decrease in pCHPKmut protein levels is not through decreased mRNA transcription in vCHPKmut/10HA-infected cells and likely through protein instability that results in degradation through the proteasome. Expression of pUS10 was also not affected at the transcriptional level between the three viruses, but the sequential decrease in pUS10 levels from vCHPKwt/10HA to vCHPKmut/10HA and vCHPKmut/10HA to vΔCHPK/10HA could not be attributed to degradation through the proteasome.

## MD induction and horizontal transmission of epitope-tagged rMDVs in chickens

Since deletion of aa 116–496 of pCHPK and epitope tagging pUS10 did not affect MDV replication in cell culture, next we determined whether epitope tagging pUS10 affected its ability to reach the FFs, induce MD, and promote horizontal spread in chickens. Since both vCHPKwt/10HA and vΔCHPK/10HA express pUL47eGFP that allows easy identification of infected FFs [2, 64, 66–70], we monitored the time for each virus to reach and replicate in FFs required for horizontal transmission in both experimental and natural (contact) infections (Fig 6A). There were no significant differences in the total number of infected birds in the experimentally infected groups; however, at 14 days pi, fewer experimentally infected chickens were positive for vΔCHPK/10HA, which was significantly different from vCHPKwt/10HA. Similar results were previously seen with vCHPKmut [23]. The time to reach the feathers was consistent with former studies examining MDV replication and transmission in Pure Columbian (PC) chickens with >75% birds positive for MDV by 21 days pi [23, 69, 71]. Consistent with the functional pCHPK requirement for horizontal transmission and natural infection, the deletion of MDV pCHPK (vΔCHPK/10HA) completely abrogated horizontal transmission. Analysis of MD incidence in experimental and natural infected chickens also confirmed pCHPK was required for transmission and induction of MD in contact chickens (Fig 6B). All chickens showing clinical signs had gross lesions following necropsy. Contact birds remaining at termination housed with vΔCHPK/10HA were uninfected based on negative qPCR results for viral DNA in the blood and the lack of anti-MDV antibodies in the serum using IFAs (S5 Fig). These results show that tagging MDV pUS10 with 2×HA at the C-terminus did not affect the ability of MDV to reach the FFs, induce MD, and horizontally transmit in chickens and further confirms MDV pCHPK is required for horizontal transmission.

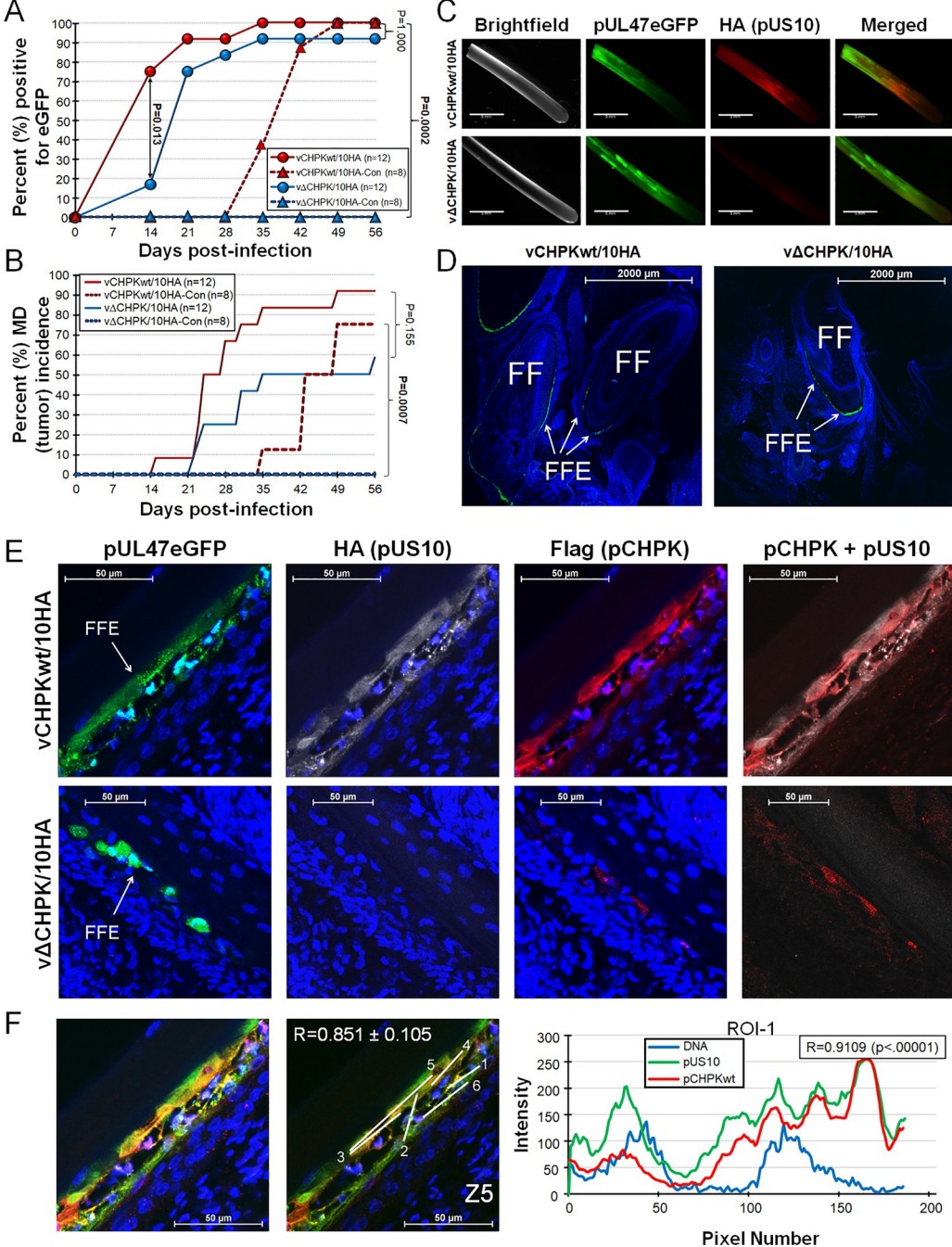

**Fig 6. Horizontal transmission, MD incidence, and pUS10 expression of HA-tagged pUS10 rMDVs in chickens.** PC chickens were infected with vCHPKwt/10HA or vΔCHPK/10HA as described in the Materials & Methods. (A) Quantitative analysis of the percent of birds with virus positive FF (pUL47eGFP) over the course of the experiment. FFs were plucked from experimentally and naturally infected groups, visualized with a fluorescent stereoscope, and determined to be positive or negative for replication in FFs based on pUL47eGFP expression. There was no significant difference between the two groups in experimentally infected chickens using Fisher's exact test ($P = 1.000$), while vΔCHPK/10HA was unable to horizontal transmit to contact (-Con) chickens that was significantly different to vCHPKwt/10HA contact chickens ($P < 0.001$). (B) Total MD incidence was determined during both experimental and natural infection. There were no significant differences in the total number of chickens developing MD in experimentally infected chickens ($P = 0.155$), while no chickens developed MD in vΔCHPK/10HA contact chickens (vΔCHPK/10HA-Con) that was significant using Fisher's exact test ($P < 0.001$). (C) Feathers were plucked from experimentally infected chickens at 28 days pi and stained for pUS10 with the anti-HA antibody showing aberrant pUS10 expression in vΔCHPK/10HA-infected feathers. Scale bars are 5 mm. (D) Representative images of skin/feather tissues cryosections cut

transversely through the FF infected with vCHPKwt/10HA or vΔCHPK/10HA-infected chickens at 30 days pi as evidenced by pUL47eGFP expression. Feather follicles (FF) and FF epithelial (FFE) skin cells are labeled in each image. Scale bars are 2000 μm. (E) High magnification of representative FFE skin cells stained with anti-HA (pUS10) and -Flag (pCHPK) antibodies. Also shown is pUL47eGFP (green) and Hoechst 33342 staining for DNA (blue). Scale bars are 50 μm. (F) Quantification of pCHPK and pUS10 in vCHPKwt/10HA-infected FFE cells. Representative image of merged Z-stacks and Z-stack with 5 linear ROIs labeled. Scale bars are 50 μm. The Pearson correlation coefficient R was measured for each ROI and averaged. Graphical presentation of the mean Pearson Correlation Coefficient R between pCHPK and pUS10 for ROI-1 is shown.

## Identification and expression of MDV pUS10 in FFE skin cells

Next, we focused on pUS10 expression in FFs and FFE skin cells. Feathers plucked from experimentally infected chickens showed pUL47eGFP expression in both vCHPKwt/10HA and vΔCHPK/10HA groups (Fig 6C). When anti-HA antibody was applied to pUL47eGFP positive FFs, no expression of pUS10 could be detected in the vΔCHPK/10HA-infected FFs. This is similar to what was observed during cell culture replication (Fig 3). To confirm this result, skin/feather cryosections were examined using confocal microscopy (Fig 6D). Similar to our former work with vCHPKmut [23], pUL47eGFP expression was noticeably lower in vΔCHPK/10HA- relative to vCHPKwt/10HA-infected FFE skin cells (Fig 6E). While pCHPK and pUS10 were abundantly detected in FFE skin cells infected with vCHPKwt/10HA, only a low level of pΔCHPK could be seen (Note: 3×Flag tag remains on the truncated pCHPK) and pUS10 was undetected in vΔCHPK/10HA-infected FFE skin cells.

Quantitation of colocalization of pCHPKwt and pUS10 in FFE cells infected with vCHPKwt/10HA produced results similar to those shown in cell culture (Fig 3E) with a strong Pearson correlation coefficient R of 0.851±0.105 (Fig 6F) using five ROIs from three cells. These data, in combination with cell culture studies, demonstrate that pCHPK and pUS10 colocalize in FFE cells, and pCHPK is required for pUS10 expression.

## Confirmation of pCHPK-mediated phosphorylation of pUS10 in FFE cells

To validate our preliminary data concerning S20 phosphorylation on pUS10 during MDV infection, we examined the global proteome and phosphoproteome of infected FFE cells. This two-pronged approach is summarized in Fig 7A. Briefly, we collected FFs from chickens infected with vCHPKwt/10HA or vΔCHPK/10HA (n = 3/group) at 21–35 days pi. Infected FFE cells were scraped from the feathers, and the proteins were extracted and digested. A fraction of each sample was used for whole proteome analysis, while the remaining peptides were enriched for phosphorylation prior to LC-MS/MS analysis. Of the numerous viral proteins phosphorylated in vCHPKwt and vΔCHPK-infected FFE cells, pUS10 was again found to be differentially phosphorylated in vCHPKwt/10HA-infected FFE cells compared to vΔCHPK/10HA-infected cells. Interestingly, we did not detect the original peptide ([R].VD**S**PKEQSY-DILSAGGEHVALLPK.[S]) from our preliminary study but identified additional and small peptides, most likely due to the implementation of an optimized tryptic digestion protocol. Now, four phosphorylated peptides were identified, exclusively in samples from vCHPKwt/10HA-infected FFE cells that were not detected in vΔCHPK/10HA-infected chickens.

Combining the data from the vCHPKwt/10HA and vΔCHPK/10HA-infected samples, pUS10 was confidently identified by 20 peptide sequences; 19 of these peptides were phospho-peptides (Fig 7B). Interestingly, there were differences in the phosphorylation pattern of pUS10 based on the presence or absence of a functional form of the pCHPK. More specifically, pUS10 was consistently phosphorylated (3/3) on residues S5, S10, S20, and S91 in the vCHPKwt/10HA-infected FFE skin cells, but never (0/3) in vΔCHPK/10HA-infected FFE skin cells. Phosphorylation of S5 and S10 do not fit CKII (SxxE/D) or CKD (SPxK) motifs, while

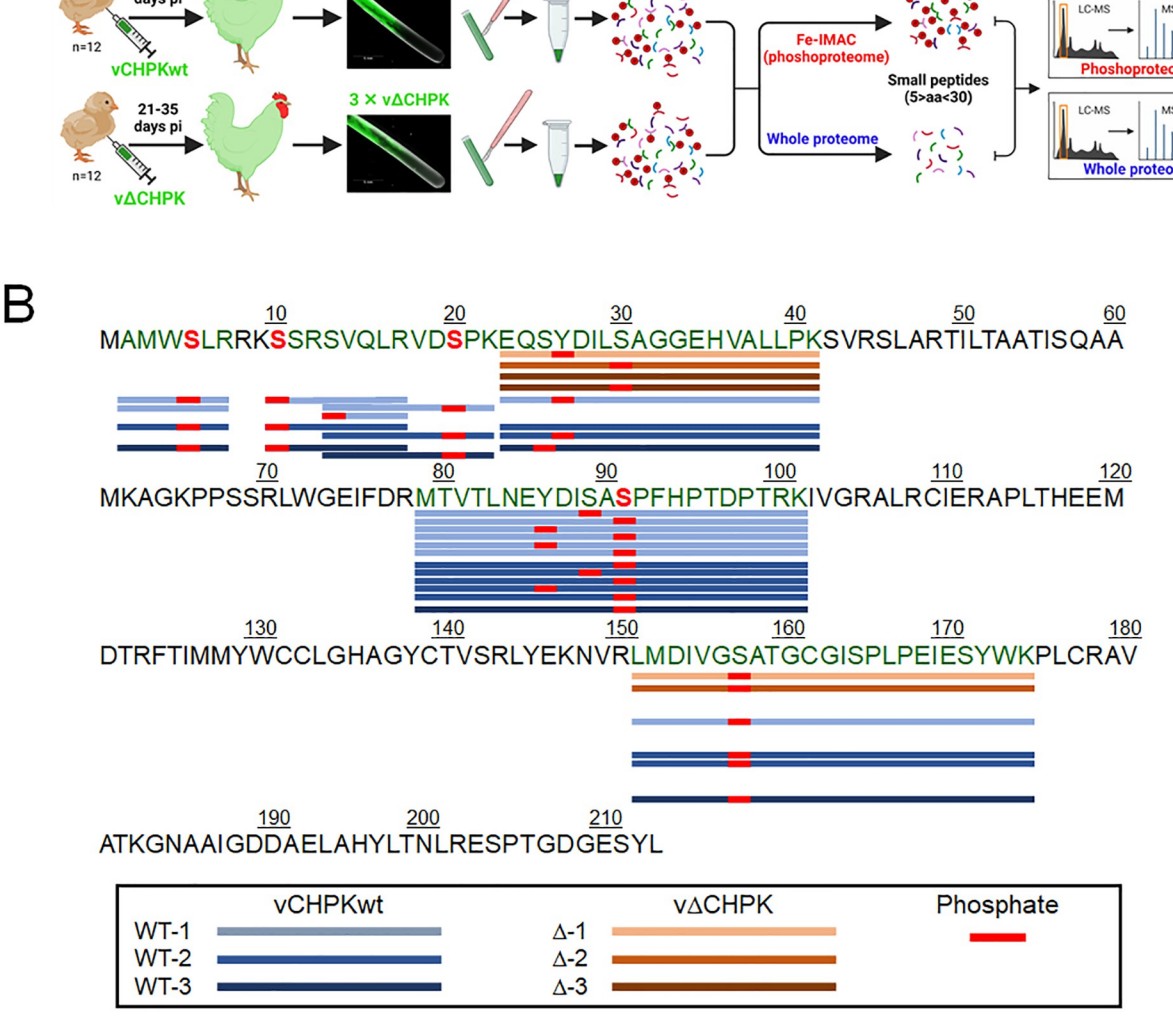

**Fig 7. pCHPK targets pUS10 in chicken epithelial skin cells.** (A) Flow chart created with BioRender.com of the experimental plan to perform MS-based proteomics on infected FFE skin cells scraped from chickens infected with vCHPKwt/10HA or vΔCHPK/10HA and prepared for LC-MS/MS-based proteomics. Some sample was used to identify the viral proteome and some protein was used for phosphopeptide enrichment. (B) The protein sequence of MDV pUS10 with peptides identified in the three replicates of vCHPKwt- (WT) and vΔCHPK- infected FFE cells that were phosphorylated. Serines phosphorylated in only vCHPKwt (3/3) and absent in vΔCHPK (0/3) samples are indicated as a red (S), including S5, S10, S20, and S91.

both S20 and S91 fit potential CKII and CKD motifs using multiple prediction programs [58, 59, 72]. A complete analysis of both cellular and viral proteins and phosphopeptide analyses of these samples will be discussed in a separate publication.

## Generation and replication of pUS10-null rMDV in cell culture

Although we did detect multiple pUS10 peptides in vΔCHPK-infected samples, pUS10 expression appeared to be completely abrogated in the absence of pCHPK expression in CECs (Fig 3B and 3C) and FFE skin cells (Fig 6E). This suggested pUS10 is dysregulated in the absence of pCHPK and may be responsible for the lack of horizontal transmission when pCHPK is mutated or deleted. MDV pUS10 had been previously implicated to be necessary

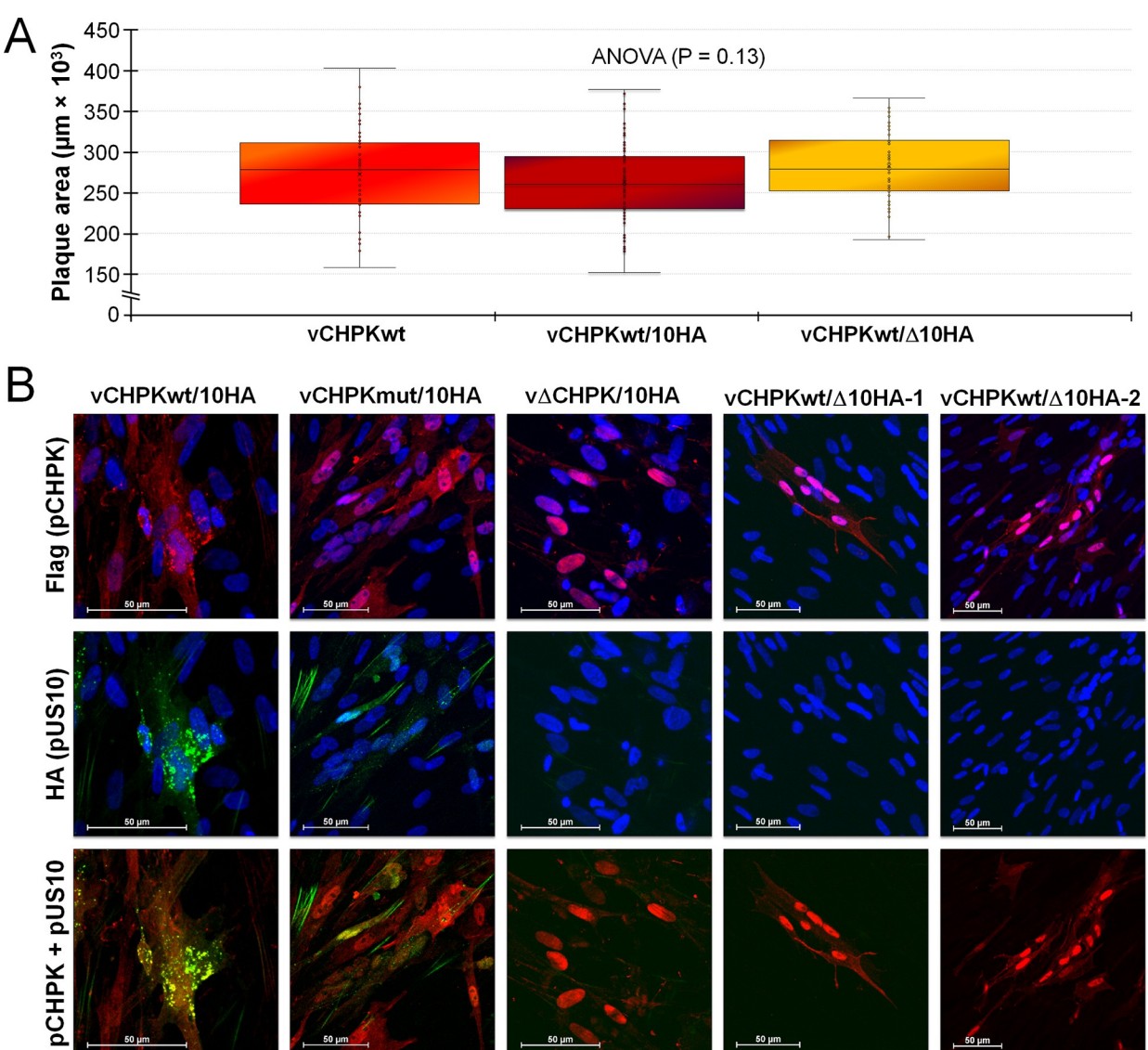

**Fig 8. Replication of vCHPKwt/Δ10HA in cell culture.** (A) Plaque areas were measured in CECs infected with vCHPKwt, vCHPKwt/10HA, or vCHPKwt/Δ10HA. There were no significant differences between all viruses using ANOVA, showing pUS10 is dispensable during cell culture replication. (B) Infected CECs were fixed and stained for pCHPK (Flag) and pUS10 (HA) using IFAs. Scale bars are 50 μm.

for horizontal transmission [57]. This information, along with our current results, strongly suggested that pUS10 is a major downstream target for pCHPK that is essential for MDV horizontal transmission. To test this hypothesis, the complete open reading frame of US10 was deleted from the MDV genome to generate rCHPKwt/Δ10HA (Fig 2C) and reconstituted into virus (vCHPKwt/Δ10HA). Fig 8A shows that removal of US10 did not (p>0.05) affect plaques sizes. IFAs confirmed no expression of pUS10 in both the vΔCHPK/10HA and vCHPKwt/Δ10HA-infected cells relative to vCHPKwt/10HA (Fig 8B). Interestingly, the localization of pCHPKwt was significantly changed in the absence of pUS10 expression in that it was mostly nuclear, similar to localization of the truncated pCHPK expressed in the vΔCHPK/10HA (Fig 3C). To be certain this localization was not a clone specific result, we also examined another independent vCHPKwt/Δ10HA clone and saw similar results (vCHPKwt/Δ10HA-2).

These data confirm deletion of US10 in vCHPKwt/Δ10HA was achieved, pUS10 is dispensable for cell culture propagation, and suggests pCHPK and pUS10 regulate expression or localization of each other in cell culture.

## Replication and horizontal transmission of pUS10-null rMDV

Next, the ability of vCHPKwt/Δ10HA to replicate in experimentally infected chickens and to infect naïve contact chickens was tested. qPCR assays for MDV genomic copies in the blood of experimentally infected chickens showed significant differences ($p < 0.05$) in virus replication at 10 and 35 days pi but no other differences at 4, 7, 14, 21, and 28 days pi between the two groups (Fig 9A). Monitoring of FF positivity for MDV and MD incidence showed both viruses reached the FFs at similar times in experimentally infected chickens (Fig 9B) and efficiently induced MD (Fig 9C). IFAs of FFs and FFEs confirmed pUS10-null MDV was negative for pUS10c2×HA (Fig 9D and 9E), while vCHPKwt/10HA abundantly expressed pUS10. However, contrary to our hypothesis that pUS10 was required for horizontal transmission, all contacts in both groups were infected by day 56 pi (Fig 9B) based on the positivity of pUL47eGFP in FFs, conclusively showing that MDV pUS10 is not required for horizontal transmission in chickens.

## Reduced MD induction by pUS10-null rMDV

There was no significant difference ($P > 0.05$) between vCHPKwt/10HA and vCHPKwt/Δ10HA for MD induction in experimentally infected chickens with both inducing 88 and 75% MD (tumor) incidence, respectively (Fig 9C). However, only 33% of contact chickens housed with vCHPKwt/Δ10HA developed MD compared to 100% contact chickens housed with vCHPKwt/10HA by 56 days pi which was statistically significant. These data suggest that, although pUS10 is not required for disease induction and tumorigenesis during experimental infection, which bypasses the natural route of infection, pUS10 may play a role in establishing infection and inducing disease during natural infection.

## pUS10 is phosphorylated in FFE skin cells

Expression of pUS10 was evident in FFs (Fig 9D) and FFE cells (Fig 9E) from chickens infected with vCHPKwt/10HA, while absent in vCHPKwt/Δ10HA-infected tissues, as expected. Since our original finding that MDV pUS10 was phosphorylated in vCHPKwt-infected FFE skin cells but not in vCHPKmut-infected FFE skin cells, we examined expression of pUS10 in FFE cells with immunoblotting. Using FFE cell protein collected from chickens infected with vCHPKmut/10HA, vCHPKwt/10HA, and vCHPKwt/Δ10HA, both unphosphorylated and phosphorylated forms of pCHPK were observed in vCHPKwt/10HA-infected FFE cells (Fig 9F). Dephosphorylation of total protein with lambda (λ) phosphatase confirmed the lower band was indeed the phosphorylated form (pCHPK*), with an abundance of pCHPK* as previously shown [23]. Also consistent with earlier studies, little phosphorylated pCHPK was seen in vCHPKmut/10HA-infected cells, and a similar result was observed in vCHPKwt/Δ10HA-infected cells.

When pUS10 expression was evaluated, two forms of pUS10 were also seen in vCHPKwt/10HA-infected cells, while only one band was observed in vCHPKmut/10HA-infected FFE cells (Fig 9F). Dephosphorylation of total protein confirmed the lower band is the phosphorylated form (pUS10*). This would be consistent with its absence in vCHPKmut/10HA-infected FFEs, as indicated in the MS-based proteomics that initiated this study. In all, these data show phosphorylation of pCHPK appears to be altered when pUS10 is not present and pCHPK is required for phosphorylation of pUS10 in FFE cells.

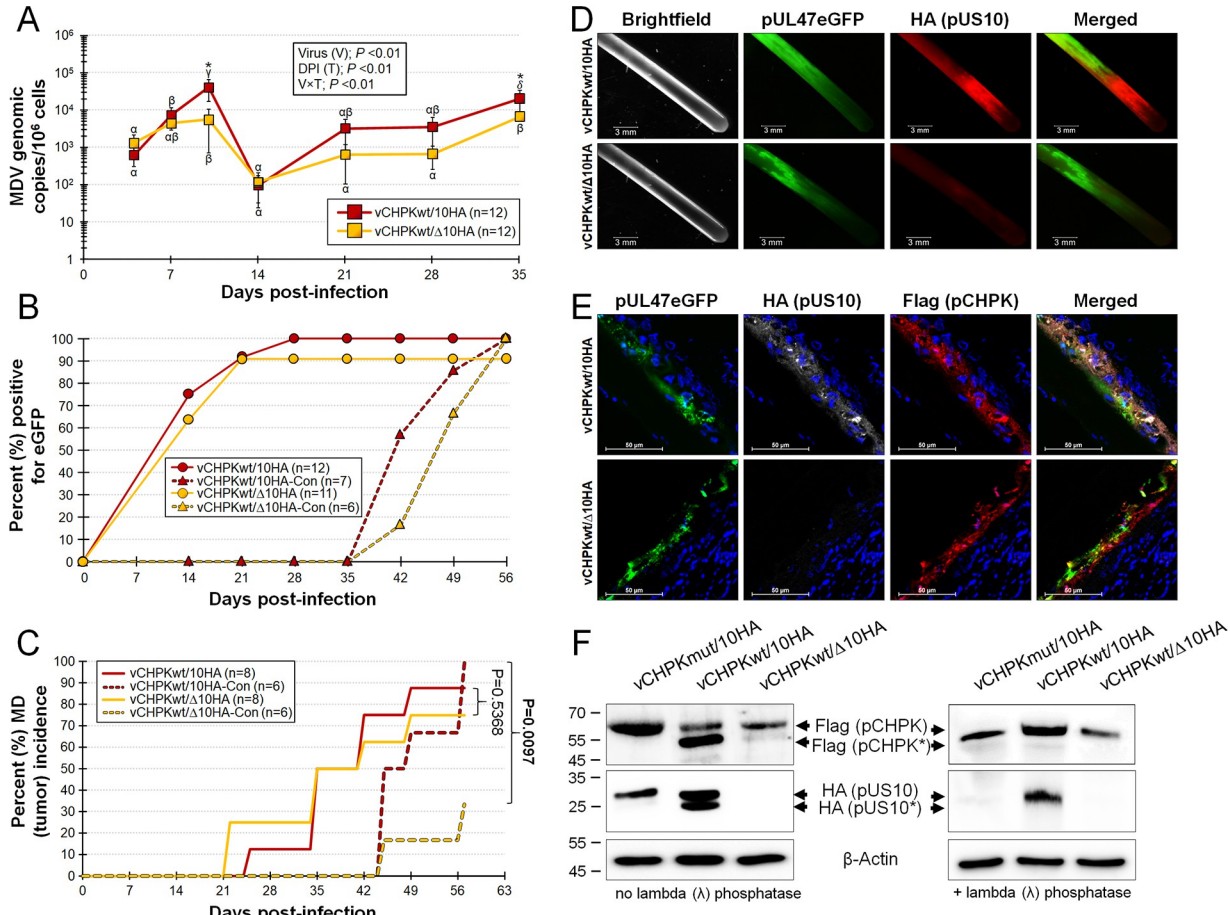

**Fig 9. Replication, MD induction, and horizontal transmission of pUS10-null rMDV.** PC chickens were infected with vCHPKwt/10HA or vCHPKwt/Δ10HA as described in the Materials & Methods. (A) Mean MDV genomic copies/$10^6$ blood cells ± standard error of means was determined in experimentally infected chickens using qPCR assays. Significant differences were determined for both virus (V) and time (T) and their interactions (V×T). Significant differences between the two groups were determined at 10- and 35-days pi (*, p <0.05). Values with Greek letters are significantly different within each group at other time points (p <0.05, two-way ANOVA, Tukey's, n = 252). (B) FFs were plucked from experimentally and naturally infected groups and the percent of birds with FF positive for MDV based on pUL47eGFP expression was determined. There were no significant differences between the two groups in experimentally- (*P* = 0.48) and naturally- (*P* = 1.00) infected chickens using Fisher's exact tests. (C) Total MD incidence was determined for each group. There were no significant differences in the total number of chickens developing MD in experimentally infected chickens, while chickens naturally infected were significantly different by day 57 dpi using Fisher's Exact Tests. (D) Feathers were plucked from experimentally infected chickens at 35 days pi and stained for pUS10 with the anti-HA antibody. Scale bars are 3 mm. (E) Feather/skin tissue cryosections cut transversely through the FFs from vCHPKwt/10HA and vCHPKwt/Δ10HA-infected chickens at 35 days pi. Sections were fixed and stained with anti-Flag (pCHPK) and -HA (pUS10) antibodies. Also shown is pUL47eGFP (green) and Hoechst 33342 staining for DNA (blue). Shown are representative images of four different FFs for each virus and antigen staining. Scale bars are 50 μm. (F) Immunoblotting under reducing conditions of total protein collected from infected FFE cells scraped from FFs. Proteins were collected from vCHPKwt/10HA and vCHPKwt/Δ10HA from the current experiment and vCHPKmut/10HA from a separate bird experiment. Anti-Flag, -HA, and β-Actin Mabs were used to identify pCHPK, pUS10, and β-Actin (protein loading control), respectively. Phosphorylated pCHPK was formerly shown to be the lower band [23] and denoted as pCHPK*, while predicted phosphorylated US10 is also denoted as such.

## Discussion

It has been firmly established that MDV pCHPK is required for horizontal transmission in chickens [2, 23, 39, 40, 42]. However, the mechanisms pCHPK mediates in this process are still not clear. Here, we showed that MD virions produced in chicken epithelial skin cells were considerably less infectious when lacking functional pCHPK. Using a LC-MS/MS-based proteomics approach to identify potential targets of pCHPK, we identified the virion protein, pUS10.

The data presented in this report showed that pUS10 expression is dependent on pCHPK and pUS10 also regulates pCHPK localization in cell culture, suggesting both genes are important for each other's proper expression. These data also suggested a major role of pCHPK during horizontal transmission was through pUS10 expression. However, when pUS10-null rMDV was tested in chickens, the virus efficiently spread from chicken to chicken, showing pUS10 is not required for MDV replication in experimentally infected chickens and horizontal transmission. However, chickens naturally infected with pUS10-null virus had less disease induction, which warrants further studies on the role pUS10 may play during natural infection and disease induction. It also emphasizes the importance of studying natural infection in the host.

Interestingly, pUS10 is not required for transmission, contrary to former studies [57], but appears to be important for the development of MD during natural infection, potentiating a role for pUS10 in establishing infection and disease induction. The reason for this could be due to several factors. First, the virus load being shed from the pUS10-null experimentally infected chickens may be lower, resulting in a delay in disease induction. There was a slight delay in positivity in contact chickens between vCHPKwt/10HA and vCHPKwt/Δ10HA (Fig 9B); however, this was not a significant difference (P = 0.2657) and unlikely to explain the significant decrease in MD incidence in contact chickens (Fig 9C). Secondly, based on expression of fluorescent viral protein in FFs in experimentally-infected chickens, there was no obvious difference in infection levels in FFs between vCHPKwt/10HA and vCHPKwt/Δ10HA (Fig 9B and 9D) that would explain the significantly reduced MD incidence at 9 weeks (Fig 9C). Thirdly, pUS10 may play a role in establishing infection during natural infection via the respiratory route. Experimental infection by inoculation of cell-associated virus bypasses the natural route of infection and can often not provide adequate information on the true pathogenesis of the disease and the role specific viral genes play, as was previously shown for other viral genes including pCHPK, gC, and pUL47 [23, 39, 40, 42, 73]. Former studies have shown MDV pUS10 to interact with cellular lymphocyte antigen 6, locus E (Ly6E) using an *E. coli* two-hybrid screening system and Ly6E was found to be associated with the length of survival and incidence of tumors following MDV challenge [74]. It is tempting to speculate pUS10 interaction with Ly6E during initiation of infection may be important and could explain the reduced MD incidence during infection via the natural route. Ly6E (formerly RIGE, SCA2, and TSA1) belongs to the LY6/uPAR superfamily that consists of proteins containing eight to ten cysteines that form a highly conserved, three-finger folding motif [75] and enhances viral uncoating after endosomal escape that is evolutionarily conserved among many species. Mar *et al.* [76] showed that Ly6E from multiple mammalian species are highly conserved and function similarly using influenza A virus as a model. Thus, interaction of pUS10 with Ly6E may enhance initiation of MDV infection in the host. Our cell-free virion infectivity data (Fig 1D) would support this hypothesis, but further studies are needed to address this potential. In all, pUS10 may play a role during horizontal transmission of MDV; however, this role is minimal when taken on its own, but likely compounded when all the potential functions pCHPK are combined, as seen in pCHPK-null virus infection (vΔCHPK), including late viral gene expression previously shown [23].

A major novel observation in this report is the co-regulation/localization of pCHPK and pUS10, as both are strongly affected by one another. Without pCHPK, pUS10 expression is nearly absent both in cell culture and in FFE skin cells *in situ*, while without pUS10, pCHPK localization is almost exclusively nuclear in cell culture (Fig 8B). In addition, the phosphorylated state of both proteins is altered in FFE cells when each gene is mutated or deleted (Fig 9F), suggesting they co-regulate each other. Normally, pCHPK expression is nucleocytoplasmic in cell culture (Figs 3C and 8B). Mutation of the invariant lysine (K170M) consistently increases nuclear localization [23], and the data here confirmed this finding (Fig 3C). Since making a complete deletion of pCHPK was not possible due to its overlapping with UL14

(Fig 2A), pΔCHPK/10HA did contain the N-terminus of pCHPK, and this truncated protein is localized almost exclusively in the nucleus. Using two different NLS prediction programs, two regions of positively charged aa clusters were predicted to contain NLS sequences that would be expressed in vΔCHPK/10HA (Fig 2D). The nuclear localization of the truncated pCHPK strongly suggests an NLS in this portion of MDV pCHPK, and regions within the deleted protein (aa 116–496) may be involved in the regulation of this localization. It is likely pUS10 is also involved in pCHPK localization since pCHPKwt was mostly nuclear in the absence of pUS10 (pCHPKwt/Δ10HA) during cell culture propagation in CECs (Fig 8B). However, localization of pCHPK in FFE skin cells in the absence of pUS10 was difficult to assess due to the structural nature of FFE skin cells. In addition, expression of the truncated pCHPK in FFE skin cells was difficult to ascertain given the low level of expression observed in these cells (Fig 6E). Analyses of other herpesviruses have suggested the N-terminal portion of some CHPKs contain NLS regulatory elements that can be regulated by cyclins [77]. Our data suggest a more detailed analysis of the putative NLS will be needed to confirm this possibility for MDV.

An important report was recently published addressing the role of phosphorylation on CHPK and stability. Iwahori *et al.*, [78] showed that binding of the regulatory proteins in the 14-3-3 family were important for stability of the human cytomegalovirus CHPK (cyclin-dependent kinase, vCDK UL97). We have consistently seen stability issues with MDV pCHPK in both cell culture and in chickens [23] when its kinase activity is abrogated (vCHPKmut). Here, we see a similar, and more dramatic, effect for pUS10 in cell culture and in chickens. Little is known about the importance of chicken 14-3-3 proteins, but further studies addressing the potential role of these regulatory proteins and MDV pCHPK, and potentially pUS10, are warranted.

The absolute requirement of pCHPK for horizontal transmission of MDV suggests a multi-functional role during natural infection. Our infectivity studies of MD virions extracted from infected chickens suggests pCHPK is important for infectivity of cell-free MDV and suggests entry into the naïve host during natural infection is the major defect of MDV lacking functional pCHPK. This report addresses one downstream viral protein in this process and its phosphorylation in a pCHPK-mediated fashion. Since pUS10 is not required for horizontal transmission (Fig 9B), the potential positive effect of phosphorylation on pUS10 functions is diminished; however, the close regulatory relationship between pCHPK and pUS10 provides an alternative hypothesis that pCHPK phosphorylation of pUS10 has an effect on pUS10 levels. The mRNA expression of pUS10 was unaffected (Fig 5A), indicating the pUS10 levels are mediated post-transcriptionally. That is, pCHPK phosphorylation of pUS10 is important for regulation of pUS10 levels in cells; however, the precise mechanism is unknown at this time, but it is most likely not mediated through degradation through the proteasome (Fig 5B).

At this point, we can only conclude pCHPK activity is important for the phosphorylation of pUS10 in FFE skin cells, but this could be either through direct pCHPK activity on pUS10 or as an intermediator of another kinase or kinase pathway that results in pUS10 phosphorylation. The role for phosphorylation of pUS10 in this function has yet to be described but Yamada et al. [49] showed both unphosphorylated and phosphorylated HSV-1 pUS10 were associated with the virion. However, as far as we are aware, this is the first description of pUS10 targeting in a pCHPK-mediated mechanism. It is important to note that only one form of pUS10 was observed in cell culture (Fig 3D) and in vCHPKmut/10HA in FFE cells (Fig 9F), while two forms, one we show is the phosphorylated form, is observed in FFE cells (Fig 9F). These data build upon our overall hypothesis that MDV pCHPK is "inactive" during cell culture propagation that helps to explain its lack of infectious cell-free virus production *in vitro*. In contrast, pCHPK is "active" in FFE skin cells *in vivo*, where fully productive virus replication is achieved, emphasizing the importance of utilizing the natural MDV-chicken model in understanding this conserved protein in herpesviral infections and pathogeneses.

In summary, our results show pCHPK and pUS10 regulate each other post-transcriptionally, and while regulation of pUS10 by pCHPK does not play a significant role during horizontal transmission of MDV in chickens, it does play a role in disease induction in naturally infected chickens. Most likely, the mechanistic importance of pCHPK for horizontal transmission of MDV in chickens is multifaceted, and pUS10 contributes to this phenotype. Very little is known about the function and role pUS10 homologs play during herpesvirus replication. HSV-1 pUS10 is a capsid/tegument-associated protein [49], suggesting it plays a role during initiation of infection. The viral gene is conserved among the *Alphaherpesvirinae* with some encoding more than one copy (*i.e.*, EHV-1 and VZV) located in the repeat short (RS) regions, while others are within the ascribed US region. Thus, it was likely translocated during divergence of the different species. For example, HSV US genes are numbered in order (US1-US12) where US1 and US10 are separated by nine US genes (2–8), while MDV US10 is located between US1 and US2 in the MDV genome. This arrangement is similar to what is seen with VZV ORF63/70 (US1-ICP22) and ORF64/69 (US10). Interesting, MDV US1 and US10 genes are believed to be expressed off a bicistronic transcript [79]. The evolutionary importance of pUS10 homologs is not known, but it has been suggested for EHV-1 pUS10 (IR5) to be associated with the viral genome in nucleocapsids, perhaps via its zinc-finger domain [51]. As far as we are aware though, no further studies have been published to address this. *In silico* analysis of MDV pUS10 did not identify a potential zinc-finger binding domain, so it would be unlikely to play similar role for MDV. However, we could confirm MDV pUS10 to interact with pCHPK in cell culture and it appears to be involved in localization of pCHPK within the cell. Further studies are required to fully address their co-operative role in gene regulation; however, our results present strong evidence for their cooperative roles during natural infection in the host.

## Materials and methods

### Ethics statement

All animal work was conducted according to national regulations. The animal care facilities and programs of UIUC meet all the requirements of the law (89–544, 91–579, 94–276) and NIH regulations on laboratory animals and are in compliance with the Animal Welfare Act, PL 279 and is accredited by the Association for Assessment and Accreditation of Laboratory Animal Care (AAALAC). All experimental procedures were conducted in compliance with approved Institutional Animal Care and Use Committee protocols. Water and food were provided *ad libitum*.

### Cell culture and cells

Specific-pathogen-free (SPF) chicken kidney cells (CKCs) were prepared from 2–4 weeks-old chickens, and CECs were prepared from 10–11-day embryos obtained from the UIUC Poultry Farm using standard methods [80], exactly as previously described [69, 70]. The chicken DF-1-Cre fibroblast cell line [81] was used to reconstitute rMDVs as previously described [68]. All cells were maintained at 38˚C in a humidified atmosphere of 5% $CO_2$.

### Recombinant (r)MDVs

The parental viruses used here were recently reported in which a 3×Flag epitope was inserted in frame of MDV UL13 at the C-terminus [23]. These included wildtype rCHPKwt and kinase mutant vCHPKmut [2, 40]. An additional CHPK mutant was generated (rΔCHPK) that only contains aa 1–115 connected to the C-terminal 3×Flag epitope to preserve the overlapping

UL14 ORF (Fig 2A). To do this, the I-*Sce*I-*aphAI* cassette was amplified by PCR from pEP--KanS2 using primers shown in S1 Table and Thermo Scientific Phusion Flash High-Fidelity PCR Master Mix. The PCR product was then used for mutagenesis in GS1783 *E. coli* cells using two-step Red recombination [82].

Following the construction of rΔCHPK, all three viruses (rCHPKwt, rCHPKmut, and rΔCHPK) were further modified by adding 2×HA tag to the C-terminus of US10 to generate double-tagged rCHPKwt/10HA, rCHPKmut/10HA, and rΔCHPK/10HA. To accomplish this, two sets of primers were designed to create the appropriate PCR product for two-Step Red recombination (S1 Table). One set of primers was used to generate a nested PCR product containing the I-*Sce*I-*aphAI* cassette from pEP-KanS2. A second set of primers was used to extend the ends to include the unique upstream and downstream integration sites. This PCR product was used for mutagenesis in GS1783 *E. coli* cells using two-step Red recombination with rCHPKwt, rCHPKmut, and rΔCHPK. Subsequently, US10 was removed from rCHPKwt/10HA to generate rCHPKwt/Δ10HA using primers in S1 Table and two-step Red recombination. Two clones were resolved from independent integrates and named vCHPKwt/Δ10HA-1 and vCHPKwt/Δ10HA-2. Only vCHPKwt/Δ10HA-1 was used in animal experiments.

All final clones were screened by RFLP (Fig 2), analytical PCR, and sequencing using previously published primers for UL13 [23] and primers shown in S2 Table. SnapGene 6.0.7 software (from Insightful Science; available at snapgene.com) was used to analysis gene sequencing results. rMDV clones were reconstituted by transfecting DF-1-Cre cells with purified BAC DNA with Lipofectamine 2000 (Invitrogen). Transfected DF-1-Cre cells were mixed with freshly prepared primary CECs and propagated in CECs until virus stocks were stored and titrated. All rMDVs were used at passage level four (p4) in all animal studies.

## Plasmids and transfections

pCHPK and pUS10 expression plasmids were produced by cloning CHPK, and its derivatives, and US10, from rMDVs in this report (Fig 2). Gibson assembly reactions were used to clone pCHPKwt, pCHPKmut, and pΔCHPK, including the 3×Flag tag, into pcDNA3.1 (Thermo Fisher Scientific, Waltham, MA). Briefly, pCHPKwt, pCHPKmut, and pΔCHPK were PCR amplified from rCHPKwt, rCHPKmut, and rΔCHPK (Fig 2) using primers shown in S3 Table and mixed with BamHI linearized pcDNA3.1 during the Gibson assembly reaction to generate pcCHPKwt, pcCHPKmut, and pcΔCHPK. US10 was cloned in frame with eGFP in the pCMS-eGFP vector (Clontech) using primers shown in S3 Table. Expression plasmids were transfected into CECs using Lipofectamine reagent according to the manufacturer's manual.

## Antibodies

The following antibodies were used according to manufacturers' instructions or previously published works. For immunoblotting, MAb H19 [83] was used to detect MDV pp38 protein as a control for the relative level of MDV infection and anti-VP5 capsid protein MAb [84] was used as a control for virion/capsid loading, and co-immunoprecipitation experiments. To detect 3×Flag tagged pCHPK and 2×HA tagged pUS10, anti-Flag M2 Mab (Sigma-Aldrich, Inc. St. Louis, MO) and rabbit polyclonal anti-HA antibody (Cell signaling Technology, Inc., Danvers, MA), respectively, were used for immunoblotting and IFAs according to the manufacturer's recommended dilution. For protein loading control in immunoblots, rabbit anti-β-Actin antibody (Abclonal, Woburn, MA) was used at its recommended dilution. Secondary anti-mouse or -rabbit IgG HRP conjugates (GE Healthcare, Piscataway, NJ) were used in immunoblotting according to the manufacturer's instructions. For IFA and plaque size assays, chicken anti-MDV polyclonal sera [85] was used. Goat anti-chicken IgY-Alexa Fluor 488 or

568 and anti-mouse or rabbit IgG-Alexa Fluor 488, 568, or Cy5 were used as secondary antibodies, and Hoechst 33342 (2 µg/ml, Molecular Probes) was used to visualize nuclei.

## Plaque size assays

Plaque areas were measured in CECs as previously described [85] using anti-MDV chicken sera and goat anti-chicken IgY-Alexa Fluor 488 or 568 secondary antibodies (Molecular Probes, Eugene, OR). Digital images of 50 individual plaques were collected using an EVOS FL Cell Imaging System (Thermo Fisher Scientific) and plaque areas were measured using ImageJ [86] version 1.53d software (http://imagej.nih.gov/ij). Box and Whisker plots were generated using Microsoft Excel 365 and significant differences were determined using IBM SPSS Statistics Version 27 software.

## RT-qPCR analysis

Total RNA was collected from 60 mm tissue culture dishes infected with 500 PFU for each virus at 5 days using RNA STAT-60 (Tel-Test, Inc., Friendswood, TX) and DNase-treated using Turbo DNA-*free* kit from Thermo Fisher Scientific using the manufacturer's instructions. RT was performed with 5–10 µg of DNase-treated total RNA using the High Capacity cDNA Reverse Transcription Kit (Thermo Fisher Scientific) and RT reactions were carried out according to the manufacturer's instructions with random primers. The reaction mixture was incubated at 25˚C for 10 min, then 37˚C for 120 min, followed by 85˚C for 5 min.

To measure cDNA levels in RT-qPCR assays, 2× Power SYBR Green PCR Master Mix (Thermo Fisher Scientific) was used with primers specific for US1, US3, US10 and UL13 (S2 Table). Primers for UL47 and UL49 have been previously published [68]. Each new primer set was designed using the NCBI/Primer-BLAST program exactly as previously done for the MDV UL47 and UL49 primers. The critical threshold ($C_T$) values were used to calculate the viral mRNA fold changes of vCHPKmut/10HA- or vΔCHPK/10HA-infected cells compared to vCHPKwt/10HA-infected cells normalized to chicken glyceraldehyde-3-phosphate dehydrogenase [68]. The expression of viral transcripts were measured in triplicate using the $2^{-\Delta\Delta Ct}$ method [87]. All RT-qPCR assays were performed using an Applied Biosystems QuantStudio 3 Real-Time PCR System (Thermo Fisher Scientific) and the results were analyzed using QuantStudio Design & Analysis Software v1.4.2 supplied by the manufacturer.

## Immunoblotting

Immunoblot analyses were performed as previously described under non-reducing (-βMercaptoethanol,-SDS) and reducing (+βMercaptoethanol,+ SDS) conditions [69]. Proteins were extracted from CECs or FFE cells in M-Per Mammalian Protein Extraction Reagent (Thermo Fisher Scientific) using the manufacturer's instructions. In some experiments, infected CECs were treated with 10 µM of the proteasome inhibitor MG132 (MilliporeSigma, Burlington, MA) or the same volume of vehicle control (DMSO) after 3 days pi and the treatment continued until protein was harvested. MAb H19 [83] was used to detect MDV pp38 protein as a control for the relative level of MDV infection or purity of MD virions. Anti-VP5 capsid protein MAb [84] was used as a control for virion loading and as a negative control for co-immunoprecipitation assays. To detect 3×Flag tagged pCHPK and 2×HA tagged pUS10, anti-Flag M2 Mab (MilliporeSigma) and rabbit polyclonal anti-HA antibody (Cell Signaling Technology, Inc.), respectively, were used according to the manufacturer's recommended dilution. For protein loading control, β-Actin rabbit Mab was used at its recommended dilution. Secondary anti-mouse or -rabbit IgG HRP conjugates were used according to the manufacturer's instructions. The SuperSignal West Pico Chemiluminescent Substrate kit from Thermo Fischer

Scientific was used to detect antigens. Protein bands were quantified using ImageJ software (version 1.6) for densitometric analysis by comparing the relative ratios of pCHPK or pUS10 to β-Actin using the technique described on the ImageJ website (https://imagej.nih.gov/ij/docs/).

## Co-Immunoprecipitation

The interaction of pCHPK with pUS10 was assessed in virus-infected cells by co-immunoprecipitation. CECs infected with vCHPKwt/10HA (2,000 PFU/ml) or mock-infected were harvested at 4 days pi. Total protein extracts were incubated with anti-HA antibody or anti-Fab (Flag) Trap agarose beads (Proteintech Group, Inc. Rosemount, IL, USA) in IP Lysis Buffer (Thermo Fisher Scientific) and protease inhibitor cocktail (MilliporeSigma) overnight at 4˚C. Anti-HA treated samples were then incubated with a pre-washed 50 μl slurry of protein A/G agarose beads (MilliporeSigma) for 2 h at 4˚C. Anti-HA and -Flag samples were then washed 4× with IP Lysis Buffer and bound proteins were eluted in Laemmli buffer. Eluted proteins and control total protein cell lysates were immunoblotted as described above using antibodies against Flag, HA, and β-Actin. An anti-VP5 antibody [84] was used as a negative control for specificity of pCHPKwt and pUS10 interactions.

## Proteomics and phosphopeptide analyses

FFE scrapings were provided to the University of Illinois Protein Sciences Facility as frozen samples. They were subsequently lysed in a buffer containing 6 M guanidine HCl, 10 mM tris (2-carboxyethyl)phosphine HCL, 40 mM 2-chloroacetamide, and 0.1% sodium deoxycholate and then boiled to promote reduction and alkylation of disulfide bonds, as previously described [88]. The samples were cleared of debris by centrifugation and subjected to chloroform-methanol precipitation to remove lipids and other impurities; the resulting protein pellets were dissolved in 100 mM triethylammonium bicarbonate with sonication. Protein amounts were determined by BCA assay (Pierce) before sequential proteolytic digestion by LysC (1:100 w/w enzyme:substrate; Wako Chemicals) for 4 h at 30˚C and trypsin (1:50 w/w; Pierce) overnight at 37˚C. Peptide samples were desalted using Sep-Pak C18 columns (Waters) and dried in a vacuum centrifuge. For phosphorylation analysis, phosphopeptides were enriched by iron-immobilized metal ion affinity chromatography (Fe-IMAC) in a microtip format before being desalted once more using StageTips [89].

Peptide samples were analyzed using a Thermo UltiMate 3000 UHPLC system coupled to a high-resolution Thermo Q Exactive HF-X mass spectrometer. Peptides were separated by reversed-phase chromatography using a 25 cm Acclaim PepMap 100 C18 column maintained at 50˚C with mobile phases of 0.1% formic acid (A) and 0.1% formic acid in 80% acetonitrile (B). A two-step linear gradient from 5% B to 35% B over the course of 110 min and 35% B to 50% B over 10 min was employed for peptide separation, followed by additional steps for column washing and equilibration. The MS was operated in a data-dependent manner where precursor scans from 350 to 1500 m/z (120,000 resolution) were followed by higher-energy collisional dissociation (HCD) of the 15 most abundant ions. MS2 scans were acquired at a resolution of 15,000 with a precursor isolation window of 1.2 m/z and a dynamic exclusion window of 60 s.

The raw LC-MS/MS data was analyzed against the Uniprot *Gallid alphaherpesvirus* 2 database (taxon 10390; 1300 sequences) using the Byonic peptide search algorithm (Protein Metrics) integrated into Proteome Discoverer 2.4 (Thermo Fisher Scientific). Optimal main search settings were initially determined with Byonic Preview (Protein Metrics) and included a peptide precursor mass tolerance of 8 ppm with fragment mass tolerance of 20 ppm. Tryptic digestion was specified with a maximum of 2 missed cleavages. Variable modifications included

oxidation/dioxidation of methionine, acetylation of protein N-termini, deamidation of asparagine, conversion of peptide N-terminal glutamic acid/glutamine to pyroglutamate, and phosphorylation of serine, threonine, and tyrosine. A static modification to account for cysteine carbamidomethylation was also added to the search. PSM false discovery rates were estimated by Byonic using a target/decoy approach, and the FDR was set to 1%.

### Preparation of cell-free viral particles

Cell-free viral particles were prepared as previously described [68] based on a combination of the Calnek [90, 91] and the Grose Laboratories' protocols [92]. Briefly, strips of skins (~100 cm$^2$) were collected from freshly sacrificed infected chickens and the feather above the calamus was clipped and removed, the subdermal fat removed, and then placed in ice-cold SPA media (7.5% sucrose in 0.01M NaPO$_4$ and 1.0% bovine serum albumin) at 5% w/v. The skins were minced with tissues in a Petri dish and then sonically disrupted at ~20 Hz for 10 s using a W-220 sonicator from Heat Systems-Ultrasonic, Inc. on ice. The samples were centrifuged at 300 $\times g$ to remove cellular debris from the supernatant. The virus particles in the clarified medium (supernatant) were pelleted by ultracentrifugation (12,000 rpm, for 2 h at 4˚C) and then gently resuspended in 1–2 ml of modified medium (Serum-free MEM without phenol red + Protease Inhibitor), stored overnight at 4˚C and layered on a 35 ml preformed continuous 5–15% Ficoll (Sigma-Aldrich, Inc.) gradient with 1 ml 25% cushion suspended in this medium. After centrifugation using a swing-out rotor (14,000 rpm for 2 h at 4˚C), the well separated particle bands were individually withdrawn by side puncture. Finally, the particles constituting the bands were pelleted by centrifugation (30,000 rpm for 1 h at 4˚C), then gently resuspended in 200–300 μl of modified medium and either used immediately or stored at—80˚C.

Two-fold dilutions of fresh cell-free viral particle extract were added to CECs in 24- or 6-well plates starting at 1:8 dilutions. Cells were incubated with diluted cell-free virus extracts for 2 h at 37˚C, after which cell-free virus extract media was removed and fresh media was added to cells. After 4–8 days, IFAs described above were used to count plaques.

### Laser scanning confocal microscopy

Infected CECs were seeded onto sterile glass coverslips in 6-well tissue culture dishes at ~500 PFU per well. At 4 dpi, cells were fixed and permeabilized with PFA buffer (2% paraformaldehyde, 0.05% Triton X-100) for 15 min, then washed twice with phosphate-buffered saline (PBS). MDV-infected skin/feather tissues were snap-frozen in Tissue Tek-optimal cutting temperature (OCT) compound (Sankura Finetek, Torrance, CA) and stored at -80˚C until sectioned. Eight- to ten-micrometer sections were affixed to Superfrost/Plus slides (Fisher Scientific, Pittsburgh, PA) and fixed with PFA buffer, then washed twice with PBS before staining with specific antibodies.

Staining with antibodies was performed as previously described [23] using A Nikon A1 Confocal Laser Microscope with the NIS-Elements C platform to capture images. All images were compiled using Adobe Photoshop version 21.0.1. Quantitative colocalization analysis was performed using ImageJ version 1.53q software on a single z-section for each image, as previously described [93]. Briefly, three cells that showed roughly equal expression levels of pCHPK derivatives and pUS10 were chosen, and five regions of interest (ROI) for each image were chosen that had clear concentrations of pUS10. A straight line was drawn that included pUS10 and a profile of pixel intensity for DNA, pUS10, and pCHPK was obtained. Pixel intensity values from each profile were used to determine a Pearson correlation coefficient using the Pearson Correlation Coefficient Calculator (https://www.socscistatistics.com/tests/pearson/) or the Data Analysis tool in Excel.

## Computational methods

The CHPK structures were modeled starting with primary sequence information using the Swiss-Model engine (https://swissmodel.expasy.org). CHPK was found to have 25.81% sequence identity with IRAK4, a transferase (PDBID:5UIQ; rcsb.org/structure/5UIQ), which was chosen as the template for modeling residues 114 to 387 of CHPK. This includes the experimental region of this study's interest, K170. The best-modeled structure from the Swiss-Model was further investigated using MDs simulations to ascertain the stability and structural movements in the presence of explicit water. AMBER18 (https://ambermd.org) tool was used to perform MDs simulations. Wildtype and mutant proteins were loaded through Xleap of AMBER18 using the AMBERff14SB force field. Appropriate number of monovalent counter ions (Cl⁻) were first added to neutralize the total formal charge on the protein, and then the protein was embedded in the center of a box of three-point water (TIP3P) molecules with a minimum distance of 12 Å between the walls to any atom in the protein. Initial coordinates and parameter files for this solvated protein were saved and the system relaxed in two steps using a non-bonded cutoff of 10 Å. In the first step, solute atoms, including the counter ions, were restrained with a force constant of 100 kcal/(mol Å$^2$). The water molecules were relaxed using 500 cycles of steepest descent and 2000 cycles of conjugate gradient method. In the second step, the whole system was relaxed using conjugate gradient minimization of 2500 cycles, including 1000 cycles of the steepest descent without any restraints. Prior to productive MDs simulations, the system was equilibrated in three phases and brought to 300 K and 1 atmospheric pressure with the integration time step of 2 fs. In the first phase, the temperature of the system was brought to 300 K using the temperature coupling with a time constant 2 ps. During the second phase, the system was brought to a constant 1 atmospheric pressure using the isotropic position scaling. Equilibration was carried out for 1 ns of total time scale with restrain 50 kcal/(mol Å$^2$) to solute. The production run was performed in NPT ensemble with the integration time step of 2 fs. Bonds involving hydrogen atoms were constrained using the SHAKE algorithm. The MDs trajectory was computed and acquired for 400 ns using our in-house GPU system. Equilibration and simulation processes were validated using the physical observables of the system. The MDs frames were recorded every 10 ps. Variations in the total, potential and kinetic energies, temperature, and pressure were estimated and further confirmed NPT properties of the system (not shown). MDs trajectory was analyzed using cpptraj tool (https://amberhub.chpc.utah.edu/cpptraj/).

## Animal experiments

Pure Columbian (PC) chickens were used for all experiments and obtained from the UIUC Poultry Farm (Urbana, IL). To avoid complications from anti-MDV maternal antibodies, chickens were experimentally infected at 3 weeks of age with 2,000 PFU of cell-associated MDV by intra-abdominal inoculation. For both animal experiments, chickens were experimentally infected with 2,000 PFU of each respective virus (n = 12/group) and housed with age-matched naïve contact chickens (n = 7-8/group) to measure horizontal transmission (natural infection).

To monitor the relative infection level in feathers, two flight feathers were plucked from each wing (4 total) weekly, starting at 14 days pi, fixed in 4% paraformaldehyde for 15 min, then washed twice with PBS. Expression of pUL47eGFP was examined using a Leica M205 FCA fluorescent stereomicroscope with a Leica DFC7000T digital color microscope camera (Leica Microsystems, Inc., Buffalo Grove, IL) as previously described [23, 69, 70]. Some heavily infected chickens, based on intense fluorescent feathers, were euthanized the next day to collect all wing feathers for protein extraction from FFE cells [94]. Briefly, fluorescent FFE skin

cells were collected from the FF using forceps and protein was collected using M-Per Mammalian Protein Extraction Reagent (Thermo Fisher Scientific) to be used in immunoblotting or frozen and processed by the University of Illinois Protein Science Laboratory for mass spectrometry (MS)-based analyses.

To measure virus replication in chickens, whole blood was obtained by wing-vein puncture, and DNA was extracted using the E.Z. 96 blood DNA kit from Omega Bio-tek, Inc. (Norcross, GA), as previously described [39]. Primers and probes against MDV ICP4 [85] and chicken iNOS [95] were used in duplex qPCR reactions in an Applied Biosystems QuantStudio 3 Real-Time PCR System (Thermo Fisher Scientific). The results were analyzed using the QuantStudio Design & Analysis Software v1.4.2. The final viral loads were presented as MDV genomic copies per cell.

To determine whether contact chickens were infected with MDV, sera was collected at the termination of animal experiments and used in IFAs. Briefly, 24-well tissue culture plates of CECs infected with MDV were fixed when plaques were evident (4–5 days pi) using PFA buffer. Sera from each chicken was diluted 1:10 and added to fixed cells for 1 h, followed by three washes with PBS. Goat anti-chicken IgY-Alexa Fluor 488 secondary antibody was added for 30 min, then washed 3 times with PBS and visualized using an EVOS FL Cell Imaging System. Chickens with antibodies against MDV will show positive staining of MDV plaques, while chickens lacking antibodies will not show positivity (S5 Fig).

## Statistical analyses

Statistical analyses were performed using IBM SPSS Statistics Version 27 software (SPSS Inc., USA) or calculators on https://www.socscistatistics.com/. Plaque size assays were analyzed with Student's *t* tests and one-way analysis of variance (ANOVA), virus was included as the fixed effect and the plaque size was used as the dependent variable. The normalized data of viral replication (qPCR) were analyzed using two-way ANOVA followed by Tukey's post hoc tests; virus (V) and time (T) and all possible interactions (V x T) were used as the fixed effects, and the genomic copies as the dependent variable. Fisher's exact test was used for infection and transmission experiments. Statistical significance was declared at $p < 0.05$, and the mean tests experiments associated with significant interaction ($p < 0.05$) were separated with Tukey's test.

## Supporting information

**S1 Fig. Sequencing results of rΔCHPK.** Sanger sequencing was performed on BAC clones using previously published primers [40]. SnapGene software was used for sequencing analysis using its alignment tool. All rΔCHPK BAC clones had the same sequence. Only shown is results for rΔCHPK.
(TIF)

**S2 Fig. Sequencing results for insertion of 2X HA epitope tag at the C-terminus of US10.** Sanger sequencing was performed on BAC clones using primers shown in S2 Table. SnapGene software was used for sequencing analysis using its alignment tool. Only shown is rCHPKwt/ 10HA but all other clones had identical sequencing results.
(TIF)

**S3 Fig. Sequencing results for deletion of US10.** Sanger sequencing was performed on BAC clones using primers shown in S2 Table. SnapGene software was used for sequencing analysis using its alignment tool. Both clones vCHPKwt/D10HA-1 and vCHPKwt/D10HA-2 are shown.
(TIF)

**S4 Fig. Predicted 3D structures from Swiss-Model and Alpha Fold.** (A-D) The predicted 3D structure from Swiss Model and Alpha fold shown in cyan and grey colors, respectively, with cationic residues of interest LYS170 and ASP293 shown in spheres. (E and F) shows the overlap of both the predicted structures with regions of interest overlapping with 4 Å RMSD. With no major change in the interacting region core structure. Note: Where B, D and F are rotated 180˚ with respect to A, C, E for better visualization of the protein region of interest.
(TIF)

**S5 Fig. Serologic test for anti-MDV antibodies from contact chickens.** Sera was tested for anti-MDV antibodies using indirect IFAs. (A) Representative images showing the results for uninfected, bird #9126 (vCHPKwt/10HA-Con), and bird #9127 (vΔCHPK/10HA-Con) as negative (-), positive (+), and negative (-) for anti-MDV antibodies. (B) The results of sera collected from uninfected or contact chickens housed with vCHPKwt/10HA or vΔCHPK/10HA-infected chickens in Fig 6B are shown.
(TIF)

**S1 Table. Primers used for generation of recombinant Marek's disease viruses.** (rMDV).
(DOCX)

**S2 Table. Primers used for sequencing, diagnostics, or RT-qPCR assays.**
(DOCX)

**S3 Table. Primers used for generation of expression plasmids.**
(DOCX)

## Acknowledgments

The authors would like to thank Dr. Yung-Tien (Yvette) Tien for her help in performing necropsy examinations during the bird experiments, as well as Dr. Widaliz Vega Rodriquez for help in holding birds during sample collections.

## Author Contributions

**Conceptualization:** Nagendraprabhu Ponnuraj, Haji Akbar, Keith W. Jarosinski.

**Data curation:** Nagendraprabhu Ponnuraj, Haji Akbar, Justine V. Arrington.

**Formal analysis:** Nagendraprabhu Ponnuraj, Haji Akbar, Justine V. Arrington, Balaji Nagarajan, Umesh R. Desai.

**Funding acquisition:** Stephen J. Spatz, Umesh R. Desai, Keith W. Jarosinski.

**Investigation:** Nagendraprabhu Ponnuraj, Haji Akbar.

**Methodology:** Nagendraprabhu Ponnuraj, Haji Akbar, Justine V. Arrington, Stephen J. Spatz, Balaji Nagarajan, Umesh R. Desai, Keith W. Jarosinski.

**Project administration:** Nagendraprabhu Ponnuraj, Keith W. Jarosinski.

**Resources:** Justine V. Arrington, Balaji Nagarajan, Umesh R. Desai, Keith W. Jarosinski.

**Supervision:** Keith W. Jarosinski.

**Validation:** Nagendraprabhu Ponnuraj, Haji Akbar, Justine V. Arrington, Keith W. Jarosinski.

**Visualization:** Nagendraprabhu Ponnuraj, Balaji Nagarajan, Umesh R. Desai, Keith W. Jarosinski.

**Writing – original draft:** Nagendraprabhu Ponnuraj, Keith W. Jarosinski.

**Writing – review & editing:** Nagendraprabhu Ponnuraj, Haji Akbar, Justine V. Arrington, Stephen J. Spatz, Balaji Nagarajan, Umesh R. Desai, Keith W. Jarosinski.

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
