## [Decision Letter · Decision Letter 0]

22 Nov 2022

Dear Dr. Jarosinski,

Thank you very much for submitting your manuscript "The alphaherpesvirus conserved pUS10 is important for natural infection and its expression is regulated by the conserved Herpesviridae protein kinase (CHPK)" for consideration at PLOS Pathogens. As with all papers reviewed by the journal, your manuscript was reviewed by members of the editorial board and by several independent reviewers. In light of the reviews (below this email), we would like to invite the resubmission of a significantly-revised version that takes into account the reviewers' comments.

The reviewers considered your manuscript as interesting and well-written, but raised several issues that would need to be addressed in any resubmission. While all reviewer comments would need to be addressed, particularly important ones include additional controls for IP, co-localization, and phosphorylation experiments, excluding secondary mutations as a source of the phenotype of mutant viruses, exploring autophagy as a possibility for the decrease in Us10 protein levels, and eliminating the use of "data not shown".

We cannot make any decision about publication until we have seen the revised manuscript and your response to the reviewers' comments. Your revised manuscript is also likely to be sent to reviewers for further evaluation.

Sincerely,

Robert F. Kalejta

Associate Editor

PLOS Pathogens

Blossom Damania

Section Editor

PLOS Pathogens

Kasturi Haldar

Editor-in-Chief

PLOS Pathogens

orcid.org/0000-0001-5065-158X

Michael Malim

Editor-in-Chief

PLOS Pathogens

orcid.org/0000-0002-7699-2064

The reviewers considered your manuscript as interesting and well-written, but raised several issues that would need to be addressed in any resubmission. While all reviewer comments would need to be addressed, particularly important ones include additional controls for IP, co-localization, and phosphorylation experiments, excluding secondary mutations as a source of the phenotype of mutant viruses, exploring autophagy as a possibility for the decrease in Us10 protein levels, and eliminating the use of "data not shown".

Reviewer's Responses to Questions

**Part I - Summary**

Reviewer #1: Ponnuraj et al., pursue the very interesting observation that the MDV conserved herpesvirus protein kinase (CHPK) is necessary for natural transmission of the virus and explore the possibility that this function is mediated by phosphorylation of one the virus tegument proteins, pUS10. They present proteomic evidence that pUS10 is phosphorylated at several specific sites in the presence of CHPK catalytic activity, but not in its absence and go on to explore the significance of CHPK and pUS10 in production of infectious virus, virus transmission and development of disease in experimentally and naturally infected chickens.

There are many interesting findings, including that the MDV CHPK catalytic activity is required for improving the specific infectivity of MDV virions, that CHPK strongly affects the expression level of pUS10 at some post-transcriptional level and that CHPK and pUS10 strongly affect each other’s localization in the infected cell. The authors also provide suggestive, but not conclusive evidence (see specific comments below), that these two proteins interact physically. The principal finding regarding the functional consequence of pUS10 phosphorylation is negative – deletion of the US10 gene has no significant effect on natural transmission but may possibly promote development of Marek’s disease following natural transmission.

1. The co-localization of pUS10 and CHPK looks very convincing in the pair of cells shown in Figure 3E, but it is not clear from the text or legend how many cells were sampled for the quantitative analysis of colocalization. This number should be at least 10 and information should be provided about how cells were selected for analysis to minimize investigator bias. This same comment applies to the colocalization results shown in Figure 5 and 6.

2. The Co-IP shown in Figure 3F is not well-controlled. Showing that VP5 is not co-precipitated goes some way toward demonstrating specificity, but important negative controls are not shown. Ideally, the results of co-IPs in which pUS10 and CHPK are not tagged should be shown.

3. The evidence that the K170M mutation changes CHPK expression is not convincing since the diminished expression of the mutant shown in Figure 3D is not at all evident in the no-inhibitor controls shown in Figure 4B. Indeed, in Figure 4B, it looks like expression of the mutant CHPK is enhanced compared to the wild-type. Since there is evidently variability in the relative expression levels, this effect would need to be carefully quantitated to be convincing. In this light, the molecular dynamics simulation of the effect of the K170M mutation in CHPK is a distraction from the rest of the experimental findings in the manuscript and should be removed.

4. The title for the legend of Figure 5 states that pCHPK and pUS10 copurify in cells, but the figure contains no copurification results.

5. The authors state that the results show that “. . . tagging MDV pUS10 with 2×HA at the C-terminus did not affect the ability of MDV to reach the FFs, induce MD, and horizontally transmit in chickens . . .” There is no condition in these experiments in which US10 is not tagged. Thus, the authors can conclude that tagging MDV pUS10 does not prevent MDV from reaching the FFs and so on, but the data leave open the possibility that tagging US10 may delay these events.

6. The authors would like to conclude that the two pUS10 bands observed in immunoblotting represent differently phosphorylated forms of the protein, since the lower band is absent in cells infected with the CHPKmut virus. This is reasonable, but not conclusive. Loss of the lower band upon phosphatase treatment is needed to demonstrate this conclusively.

7. The first and second paragraphs of the discussion repeat almost the same information.

8. The discussion does not address the large difference in specific infectivity in virions made by CHPK and CHPKmut viruses. It would be nice to know what the authors think about this and whether there might be a role for the tegument protein pUS10 in this function of CHPK.

Reviewer #2: In this clearly written article by Ponnuraj et al from the Jarolinski Group, the role of the Mareks disease virus conserved herpesvirus protein kinase (CHPK) and one of its potential targets, the US10 gene, are examined in the context of infected cells and in spread of the virus in chickens. The authors go through a number of experiments to show that the protein kinase has an influence on the amount of US10 protein in cell culture and in host animals. They have previously partially detailed the role of the CHPK previously in a publication using a virus with point mutant inactivated protein kinase (ref 23), showing that it has such a role. Here they make a new mutant deleted for a large part of the protein kinase. The identify US10 as a substrate by phosphoproteomics, and show that its phosphorylation is dependent on the viral protein kinase. It interacts with the protein kinase. Virus lacking the US10 protein shows viability and some attenuation in animal models but is surprisingly not that severe. An interesting observation is that lacking US10 changes the cellular distribution of the protein kinase. While overall the work appears to be well done, controlled and has some very intriguing observations, there are some outstanding and unresolved issues that do not outline with clarity the mechanisms going on.

Reviewer #3: In this manuscript, the authors investigated the targets of the conserved herpesvirius protein kinases (CHPK) of Marek’s disease virus (MDV) using LC-MS/MS. They identified the US10 capsid/tegument-associated phosphoprotein as a target of CHPK in MDV infection in vivo. Expression of US10 was shown to be dependent on the CHPK activity. In addition, they examined the role of US10 in this model of virus induced tumor formation and could demonstrate that the protein is dispensable for transmission but plays an important role in the disease induction in naturally infected chickens. The manuscript is very well written and contains exciting data on these two conserved herpesvirus proteins that will be of interest to herpesvirologists in general and beyond. This natural virus-host model provides intriguing data that could not have been obtained for other herpesviruses. Only the following points should be addressed prior to publication.

**Part II – Major Issues: Key Experiments Required for Acceptance**

Reviewer #1: No-tag control for Co-IP shown in Figure 3F.

No-tag control for US10 arrival at follicles, transmission and disease development.

Phosphatase treatment of pUS10 to show multiple bands are due to phosphorylation

Reviewer #2: It is not well resolved why intracellular levels of the US10 protein decline with virus point mutated in the CHPK, and why they are nearly non-existent in the CHPK deleted virus. It is apparently not proteosome mediated degradation (not affected by MG132) or at the transcriptional level, but seems to be by another mechanism. This is speculated about in the discussion, but is rather an unfinished. Does stability measured by a protein pulse chase approach show increased degradation of US10 with the lack of CHPK? It may not be just the proteosome, but rather another means of turnover (autophagy). Is the protein being made? The CHPK could be targetting the translation machinery to affect translational levels. How does expression of other late genes become affected? A wider protein study is needed.

Regarding the importance of US10, which is stated as being important for disease induction in chickens, the lack of US10 also affects the cellular localization of the Wt CHPK in cells. As such there are many potentials for lacking the US10 to be the cause of the modest attenuation of the virus in vivo. Thus the role of the US10 with the CHPK is complex and not resolved. It is not clear if the CHPK directly phosphorylates the Us10 or acts through another (cellular?) kinase… also please provide a bit more info on what US10 is in related viruses. Any suggestion of function? Is there a HSV equivalent? The statement made in line 392 is far too strong given the lack of the resolution of the relationship of CHPK and US10.

In figure 1, virions are analyzed. However only antyibody blots are shown. It is critical to show a total protein stain of the viruses so one can judge the purity of the virus preparations and the relative abundances of the capsid proteins to other virus proteins, and the consequences of CHPK mutation on the virion protein profile. A blot of a non-structural protein should be shown, which would not be in the virions. And Why is there no major capsid protein seen in the WT infected cell extracts? This is not explained and does not make sense.

Reviewer #3: 1) The authors generated an exciting set of recombinant viruses for their study. Unfortunately, they did not exclude the presence of secondary mutations in these mutants. This would be particularly important for the US10 mutant that provided very important data on the role of this protein in a natural infection. The authors (e.g. Dr. Spatz) should sequence at least this crucial mutant to ensure that this exciting phenotype is not due to a secondary mutation that can occur during mutagenesis.

**Part III – Minor Issues: Editorial and Data Presentation Modifications**

Reviewer #1: Legend for Figure 6A should specify abbreviations. After a minute of confusion, I figured out that the -Con suffix denoted the results for chicken infected by contact with experimentally infected chickens, but this information should be in the legend.

Reviewer #2: There are many sections that contain statements with data not shown. This does not conform with PLOS policy. Either the data is shown or the statement must be removed

A lot of the first part of the work seems to be confirmatory of that published in reference 23, where it was shown that the CHPK influenced the in vivo spread and replication of MDV. This reduces importance and novelty of the findings presented here. By the looks of it, figure 1A largely is repetitive of that seen in publication 23. Please specify clearly which work is not precedented and what repeats previously published work.

166-170 details the BAC analyses. However it would be helpful to add the restriction sites for the regions concerned to the figure

Line 182 please show the single fluor channels as the nuclear location is not obvious in the figure

Have the authors considered making a virus that has the US10 protein that simply lacks the sites of phosphorylation (esp the SPKE mutated to alanine, and seen if this virus is of the same phenotypes as the US10 deletion?

Figure 4 a needs a normalizing gene expected to be expressed by both viruses. could rt pcr signal for either gene come from a run on transcript from either side? This is very common in the herpesviruses. The RNA seq data not shown (in a different paper? ) could help explain the transcription of these genes without the functional CHPK.

Figure 5A it is clear to this reader that the magnification of CHPK mutation is different from that of the WT and deletion mutants. Either that, or the mutation has a consequence on nuclei size!!!! And only a single infected cell is shown. Yet the cellular distribution patters seem quite variable in infected cell culture….

Did the authors look autophpagy as a basis for the loss of ORF10

There was no other mention of other targets for the CHPK in the phosphoproteome analyses studies……

Reviewer #3: 1) Line 177ff & Figure 3: The authors identify that US10 expression severely impaired (Fig. 3B). But instead of continuing with this aspect, they jump to the localization of CHPV and its mutant (Fig. 3C). Only then they come back to the US10 expression validating it with a western blot (Fig. 3D). To keep the story line straight, the authors could optimally show the western blot earlier as 3C before addressing the CHPV localization.

2) The authors use the Swiss-Model engine to predict the structure of the CHPK as no crystal structure is available. They should reevaluate their prediction with AlphaFold 2 as this has a much greater accuracy with its AI-based predictions.

3) In their in vivo experiments, they only show the disease incidence, but not the tumor incidence. It would be great if they could provide this data in their manuscript, as they certainly have it. At least they should state if all animals that developed disease also had tumors.

PLOS authors have the option to publish the peer review history of their article (what does this mean?). If published, this will include your full peer review and any attached files.

Reviewer #1: No

Reviewer #2: No

Reviewer #3: **Yes: **Benedikt Kaufer
---

## [Decision Letter · Decision Letter 1]

30 Jan 2023

Dear Dr. Jarosinski,

We are pleased to inform you that your manuscript 'The alphaherpesvirus conserved pUS10 is important for natural infection and its expression is regulated by the conserved Herpesviridae protein kinase (CHPK)' has been provisionally accepted for publication in PLOS Pathogens.

Best regards,

Robert F. Kalejta

Academic Editor

PLOS Pathogens

Blossom Damania

Section Editor

PLOS Pathogens

Kasturi Haldar

Editor-in-Chief

PLOS Pathogens

orcid.org/0000-0001-5065-158X

Michael Malim

Editor-in-Chief

PLOS Pathogens

orcid.org/0000-0002-7699-2064

Reviewer Comments (if any, and for reference):

Reviewer's Responses to Questions

**Part I - Summary**

Reviewer #1: The revised manuscript of Ponnuraj et al., includes important new data and controls including phosphatase treatment to prove phosphorylation of viral proteins. They ahem also provided important standards of comparison from the literature and expanded the discussion as requested. All of my previous reservations have been satisfied.

Reviewer #2: good response to most of the three reviewers comments overall

Reviewer #3: In this manuscript, the authors investigated the targets of the conserved herpesvirius protein kinases (CHPK) of Marek’s disease virus (MDV) using LC-MS/MS. They identified the US10 capsid/tegument-associated phosphoprotein as a target of CHPK in MDV infection in vivo. Expression of US10 was shown to be dependent on the CHPK activity. In addition, they examined the role of US10 in this model of virus induced tumor formation and could demonstrate that the protein is dispensable for transmission but plays an important role in the disease induction in naturally infected chickens. The manuscript is very well written and contains exciting data on these two conserved herpesvirus proteins that will be of interest to herpesvirologists in general and beyond. This natural virus-host model provides intriguing data that could not have been obtained for other herpesviruses. In this resubmission of the manuscript PPATHOGENS-D-22-01876, the authors addressed many comments of the reviewers and improved the manuscript, which could now be published.

**Part II – Major Issues: Key Experiments Required for Acceptance**

Reviewer #1: None

Reviewer #2: am pleased with the response to the previous reviews. im am still a little concerned about the purified virions preparations and the lack of a response to a request for a protein stain of the virions to be convincing that they are indeed virions and not some other intracellular substructure or secreted elements, but i can sympathize that this is probably difficult to do. It still leaves a bit of a concern, but i will abide by the editor on this issue if they think a protein stain is not nescessary

my only requirement is line 225, it should be figure 5A not 4A

Reviewer #3: none

**Part III – Minor Issues: Editorial and Data Presentation Modifications**

Reviewer #1: None

Reviewer #2: (No Response)

Reviewer #3: none

PLOS authors have the option to publish the peer review history of their article (what does this mean?). If published, this will include your full peer review and any attached files.

Reviewer #1: No

Reviewer #2: **Yes: **PAUL R KINCHINGTON

Reviewer #3: No

---

## [Editor Report · Acceptance letter]

2 Feb 2023

Dear Dr. Jarosinski,

We are delighted to inform you that your manuscript, "The alphaherpesvirus conserved pUS10 is important for natural infection and its expression is regulated by the conserved *Herpesviridae* protein kinase (CHPK)," has been formally accepted for publication in PLOS Pathogens.

Best regards,

Kasturi Haldar

Editor-in-Chief

PLOS Pathogens

orcid.org/0000-0001-5065-158X

Michael Malim

Editor-in-Chief

PLOS Pathogens

orcid.org/0000-0002-7699-2064